# Probing SARS-CoV-2 membrane binding peptide via single-molecule AFM-based force spectroscopy

Qingrong Zhang[1], Raissa S. L. Rosa[2], Ankita Ray [1], Kimberley Durlet[1], Gol Mohammad Dorrazehi [1], Rafael C. Bernardi [2,3] ✉ & David Alsteens [1,4] ✉

The SARS-CoV-2 spike protein's membrane-binding domain bridges the viral and host cell membrane, a critical step in triggering membrane fusion. Here, we investigate how the SARS-CoV-2 spike protein interacts with host cell membranes, focusing on a membrane-binding peptide (MBP) located near the TMPRSS2 cleavage site. Through in vitro and computational studies, we examine both primed (TMPRSS2-cleaved) and unprimed versions of the MBP, as well as the influence of its conserved disulfide bridge on membrane binding. Our results show that the MBP preferentially associates with cholesterol-rich membranes, and we find that cholesterol depletion significantly reduces viral infectivity. Furthermore, we observe that the disulfide bridge stabilizes the MBP's interaction with the membrane, suggesting a structural role in viral entry. Together, these findings highlight the importance of membrane composition and peptide structure in SARS-CoV-2 infectivity and suggest that targeting the disulfide bridge could provide a therapeutic strategy against infection.

Coronavirus disease 2019 (COVID-19) is a global pandemic and a serious public health threat caused by the severe acute respiratory syndrome coronavirus 2 (SARS-CoV-2). SARS-CoV-2 belongs to the coronavirus family and is an enveloped positive-stranded RNA virus[1]. The surface of SARS-CoV-2 is decorated with the glycosylated homotrimer Spike (S) proteins, which play a crucial role in binding and fusion with the host cell membrane. The S protein is divided into two functional subunits, S1 and S2, through cleavage by the host protease furin in the Golgi apparatus during viral maturation[2–4]. S1 is responsible for binding to the host cell receptor, angiotensin-converting enzyme 2 (ACE2), while S2 facilitates membrane fusion[5–7]. Upon binding of the receptor-binding domain (RBD) present in the S1 subunit to ACE2, the S2′ site in the S2 subunit undergoes processing by the transmembrane protease serine 2 (TMPRSS2), leading to the shedding of S1 and exposure of the membrane binding peptide (MBP)[8]. Subsequently, the S2 subunit undergoes conformational changes, to promote MBP

binding to the host cell membrane. Finally, irreversible refolding of the S2 subunit brings the viral and host cellular membranes in close spatial proximity, leading to the membrane fusion between the viral envelope and target cells (Fig. 1a)[9,10].

Due to the crucial role of the S1 subunit in receptor recognition, current SARS-CoV-2 vaccines and therapies primarily target the S1 domain of the S protein[11–15]. However, an alternative approach is to focus on the S2 subunit, which is more genetically conserved across coronaviruses[16–19]. The SARS-CoV-2 MBP, consists of 40 amino acids (residues 816-855 of the S protein sequence), and contains a fully conserved internal disulfide bond (Cys840 and Cys851)[20–22]. Targeting this domain holds promise in addressing emerging SARS-CoV-2 variants and mitigating potential future coronavirus outbreaks. Therefore, gaining a comprehensive understanding of the molecular mechanisms governing SARS-CoV-2 MBP early binding to lipid membrane is crucial for advancing our knowledge of coronavirus infection

[1]Louvain Institute of Biomolecular Science and Technology, Université catholique de Louvain, Croix du sud 4-5, L7.07.07, Louvain-la-Neuve, Belgium. [2]Department of Chemistry and Biochemistry, Auburn University, Auburn, AL, USA. [3]Department of Physics, Auburn University, Auburn, AL, USA. [4]WELBIO department, WEL Research Institute, Avenue Pasteur, 6, Wavre, Belgium. ✉e-mail: rcbernardi@auburn.edu; david.alsteens@uclouvain.be

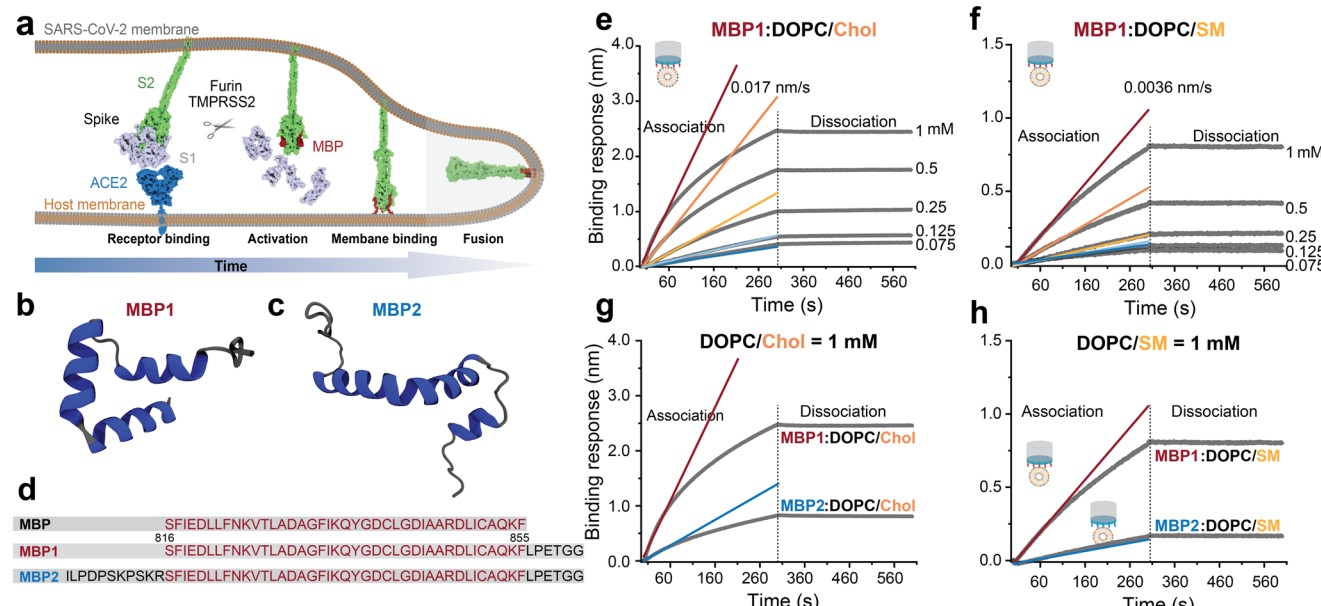

**Fig. 1 | SARS-CoV-2 MBP interacts with lipid vesicles. a** Cartoon illustrating the interaction between SARS-CoV-2 and the host membrane. The S protein binds to the host cell surface via ACE2 receptor, and upon host proteases activation, the S1 subunit detaches, exposing the MBP. The exposed MBP then binds to the host membrane, followed by refolding of S2 into a post-fusion state, leading to the membrane fusion between virus and target cells. **b, c** Ribbon diagrams of MBP1 and MBP2 respectively, predicted by AlphaFold2 and equilibrated using a molecular dynamics protocol. **d** The sequence of the MBP region of SARS-CoV-2 spans amino acid residues 816-855 of the S protein sequence. MBP1 and MBP2 are tagged with LPETGG at the C-terminus, and MBP2 is extended by an additional 11 amino acid residues at the N-terminus, mimicking the uncleaved peptide. Biolayer interferometry sensorgrams depicting the real-time binding of SARS-CoV-2 MBP and lipid vesicles consisting of DOPC and Chol (**e**) or DOPC and SM (**f**). Grey lines represent the raw data, while colored lines represent the binding rate of the association phase, which is the linear fit within the preliminary 60 s of the association phases. Biolayer interferometry sensorgrams show the comparison of the binding of MBP1 or MBP2 to the lipid vesicles. MBP1 or MBP2 (10 μM) were loaded onto the sensors and were then allowed to interact with the 1 mM DOPC/Chol vesicle (**g**) or DOPC/SM vesicle (**h**), respectively. Source data for panels e-h are provided as a Source Data file.

and facilitating the design of effective vaccines and therapies. Recent studies have demonstrated the broad efficacy of lipopeptides targeting the conserved membrane fusion process and monoclonal antibodies recognizing the conserved MBP motif in combating coronavirus infection[23–25].

Stable membrane binding provides a foothold for viral membrane fusion, an energetically favorable process that faces significant kinetic barriers due to repulsive hydration forces between lipid bilayers[26]. Notably, the height of this activation energy barrier varies substantially depending on the composition of the membranes[19,27]. In the case of SARS-CoV-2, the energy needed to overcome the barrier between the virus and the host cell membrane arises from an irreversible conformational transition in the S2 subunit. This transition converts S2 from a less stable pre-fusion state to a more stable post-fusion state[9,10], and it relies on the binding stabilization of MBP region to the membrane. The internal disulfide bond in the MBP most presumably aids in stabilizing its structure during binding. Furthermore, there is increasing evidence that the lipid composition of viral and cellular membranes[28,29], as well as the depletion of cholesterol (Chol), impact SARS-CoV-2 membrane fusion and infection[30]. However, formal evidence regarding how the internal disulfide bond of the MBP region and lipid composition contribute to MBP binding to the host membrane is still lacking.

In this work, we investigate at the single molecule level the effect of lipid composition, MBP sequence and internal disulfide bond on binding kinetics and stability upon interaction with the host membrane. Utilizing biolayer interferometry (BLI) assays, we assessed how membrane composition influences the binding kinetics of SARS-CoV-2 MBP to lipid vesicles. Subsequently, by employing atomic force microscopy (AFM)-based single-molecule force spectroscopy (SMFS), we quantified kinetic and thermodynamic parameters relevant to this interaction[31]. To validate our experimental findings, unbiased molecular dynamics (MD) simulations were conducted. Our comprehensive analysis revealed a distinct preference of MBP for cholesterol-enriched membranes over sphingomyelin-containing ones and pointed out the key role of the Arg846 in the early interaction with the lipid bilayer. Importantly, our experiments also showed that the presence of an internal disulfide bond within MBP enhances its binding to host membranes, subsequently prolonging peptide-membrane interactions that increase the overall infectivity of SARS-CoV2. These observations were further substantiated under physiologically relevant conditions, demonstrating that Chol facilitated binding between the TMPRSS2-cleaved S2 subunit and the membrane, leading to the inhibition of SARS-CoV-2 infectivity in relevant assays.

## Results

### Cholesterol facilitates the binding of SARS-CoV-2 MBP to lipid vesicles

The functional MBP of SARS-CoV-2 is a 40 amino acids sequence starting at Ser816 that is formed from the mature N-terminus of the S2′ site, after enzymatic priming by TMPRSS2[8,32]. This peptide is highly conserved across coronaviruses, and targeted by several broadly neutralizing antibodies[23,25]. To probe the dynamics of the SARS-CoV-2 MBP binding to the lipid bilayer, we designed two peptides harboring slight modifications, MBP1 and MBP2, based on the primary MBP sequence. MBP1 is the LPETGG-tagged version of MBP, allowing the protein to be ligated by sortase A at the C-terminus, and MBP2 represents the uncleaved version of MBP1, with an additional stretch of 11 amino acid residues at the N-terminus (Fig. 1d). To assess the influence of lipid composition on MBP domain binding, lipidic vesicles composed of 70% 1,2-dioleoyl-sn-glycero-3-phosphocholine (DOPC) and 30% sphingomyelin (SM) or Chol were used, mimicking the physiological composition of cell membranes[33,34].

To assess the impact of Chol on the membrane-binding capacity of SARS-CoV-2 MBP1, we conducted a biolayer interferometry (BLI) assay[35]. The MBPs were covalently attached to an amine-reactive biosensor surface. Briefly, sensors bearing free carboxylic acid groups were activated by reaction with EDC (1-Ethyl-3-[3-dimethylaminopropyl] carbodiimide hydrochloride) and NHS (N-hydroxysulfosuccinimide) to generate highly reactive NHS esters. The esters rapidly react with the amine group of MBP1 or MBP2, forming a highly stable amide bond. Subsequently, the MBP-coated biosensors were made to interact with lipid vesicles containing 70% DOPC with 30% SM or Chol in a concentration-dependent manner (0.075, 0.125, 0.25, 0.5, and 1 mM respectively) to construct the respective binding isotherms. Our findings revealed a fivefold faster binding of MBP1 for cholesterol-containing vesicles compared to sphingomyelin-containing vesicles. Specifically, the binding rate of MBP1 to DOPC/Chol (70:30) vesicles ($0.017$ nm s$^{-1}$) was five times higher than to DOPC/SM (70:30) vesicles ($0.0036$ nm s$^{-1}$) during the association phase (Fig. 1e, f). To extend the interaction to more complex lipid compositions, we tested the binding rate of MBP1 to another mixed lipid composition (DOPC/DPPC/Chol = 40:40:20). We found that the binding rate of MBP1 to DOPC/DPPC/Chol (40:40:20) vesicles is 0.018 nm/s (Supplementary Fig. 1), similar to the rate observed for DOPC/Chol (70:30) vesicles (0.017 nm/s). This consistency underscores the predominant role of Chol in enhancing binding kinetics, rather than the contributions from other lipid components. As control experiments, we also evaluated the membrane-binding response of SARS-CoV-2 MBP2 to both Chol- and SM-containing vesicles. Interestingly, we observed a significantly lower membrane-binding response of MBP2 compared to MBP1 when exposed to vesicles at the same concentration (Fig. 1g, h). These findings emphasize the significance of the N-terminus exposure at the S2' site during the membrane-binding of SARS-CoV-2 MBP. Our BLI findings highlight the critical role of Chol in facilitating MBP binding to the lipid membrane.

## Probing SARS-CoV-2 MBP-binding at the single-molecule level

While BLI measures the overall interaction between a large population of molecules, providing averaged kinetic data such as association and dissociation rates, AFM allows for the detailed analysis of individual molecular interactions, capturing subtle variations in kinetics and thermodynamics that might be obscured in ensemble measurements. Hence, to account for potential influences arising from factors such as vesicle size, coupling of MBP to the sensor and its accessibility for the vesicle, we performed an exhaustive analysis of these interactions at the single-molecule level by AFM, between oriented MBP onto the AFM tip and lipid bilayers. Supported lipid bilayers, with a similar composition as the vesicles used for BLI experiments, were prepared to mimic the outer leaflet of cell membranes[36]. To validate the formation of lipid bilayers, AFM topography imaging was performed under physiological conditions which maintain their near-native state of hydration. Height cross-section analysis revealed a height of approximately $4.74 \pm 0.1$ nm for both the DOPC/Chol and DOPC/SM systems (Fig. 2a and Supplementary Fig. 2). The integration of smaller cholesterol molecules or long-saturated acyl chain sphingomyelins (SMs) into the DOPC matrix resulted in a slightly increased bilayer thickness to approximately $4.84 \pm 0.2$ nm for DOPC/SM compared to DOPC/Chol. To reflect the physiologically relevant state of SARS-CoV-2 interaction with host cells, MBPs were covalently attached in an oriented manner to the AFM cantilever tip using a flexible polyethylene glycol spacer (PEG$_{24}$) and the Sortase A enzyme (Supplementary Fig. 3a). This approach allows for precise control over the grafting geometry, providing a reliable method to specifically probe the interaction at this site[37,38]. This ensures that the experimental setup accurately captures the movement of the exposed N terminus of the MBP and faithfully represents the encounter between SARS-CoV-2 and host cells.

To investigate the binding dynamics of the MBP, AFM-based SMFS was conducted in the height-clamp mode[39–41]. The functionalized AFM tip was positioned 4–6 nm above the lipid bilayer, allowing specific interactions between the MBP and the lipid layer to be captured as Force vs. Time (FT) curves (see Supplementary Movie 1). The resulting force (in pN) between the functionalized tip and the lipid bilayer was continuously monitored over a 10-second period, during which the MBP and the lipid bilayer interacted. Analysis of the FT curves for MBP1 revealed distinct binding events, characterized by a jump to negative forces, observed in approximately 7% of the total cases ($n = 180/2,519$ for MBP1-DOPC/Chol and $n = 206/2,851$ for MBP1-DOPC/SM; Fig. 2b, and Supplementary Fig. 4). In contrast, for MBP2, only a few binding events (<0.7% of total cases; 10/1,359 for MBP2-DOPC/Chol and $n = 6/3528$ for MBP2-DOPC/SM were detected. This experimental finding shows that the binding frequencies (BFs) remained consistent regardless of the presence of cholesterol but the enhanced infectivity is rather driven by higher forces and subsequently longer bond lifetimes after binding. This suggests that Chol does not alter the frequency of binding events, but it enhances the stability and strength of the interactions between the virus and the host cell membrane, leading to more effective viral entry and infection. As a positive control, a membrane-penetrating peptide (MPP) of the S protein (residues 866–910), validated by cryo-EM as the region that crosses the host membrane[42], binds to the DOPC/Chol membrane with an approximately 16.5% binding frequency (Supplementary Fig. 4). These findings suggest that the N-terminal priming of the MBP by TMPRSS2 significantly influences its binding properties to lipids, as evidenced by the lower binding events observed for MBP2 compared to MBP1.

The binding events within the collected FT trajectories were identified using a specialized algorithm called the generalized step transition and state identification (STaSI, see methods, section 'Analysis of FT curves and extraction of kinetic parameters'), specifically designed for analyzing discrete single-molecule data[43]. In the FT curves, a rapid jump to the negative regime was observed, indicating the transition of the MBP from an unbound state to a bound state (Fig. 2c, d). These events have a lifetime in the range of a few hundred milliseconds. Interestingly, upon initial observation, the lifetime of the MBP1 bound state appeared longer for the DOPC/Chol (~426 ms) lipid bilayer compared to the DOPC/SM (~126 ms) lipid bilayer. Despite the dependence of lifetime on lipid composition, the magnitude of force was found to be unaffected. This observation suggests that the presence of cholesterol (Chol) enhances the kinetics of SARS-CoV-2 MBP binding to the lipid membrane (Fig. 2c, d).

## Towards a quantitative characterization of MBP binding

Subsequently, the lifetime and strength of the binding events between MBP1 and lipid bilayers were determined for the two different lipid compositions (Fig. 3a). The dependencies of the lifetime on the binding strength for both lipid compositions show an exponential decay and were fitted using the Bell model[39,44]. (Fig. 3b, c). This model allows us to extract the average lifetime ($\tau(0)$) of the complex formed between MBP and lipid bilayer, as well as to estimate the width of the free-energy barrier ($x_\beta$) separating the unbound and the bound state. These results provide mechanistic insights into the kinetics of molecular complexes formed under conditions of thermal equilibrium, that is without the application of any external force.

For MBP1 binding to the DOPC/Chol membrane (Fig. 3b), the obtained $\tau(0)$ and $x_\beta$ values were $426.2 \pm 76.5$ ms and $0.29 \pm 0.04$ nm, respectively. These values were significantly higher compared to the DOPC/SM membrane (Fig. 3c), where $\tau(0)$ and $x_\beta$ were $126.8 \pm 18.9$ ms and $0.18 \pm 0.03$ nm, respectively. These findings indicate that Chol aids in the formation of a more stable lipid: MBP complex, concomitantly facilitating the further insertion of S2 into the lipid bilayer.

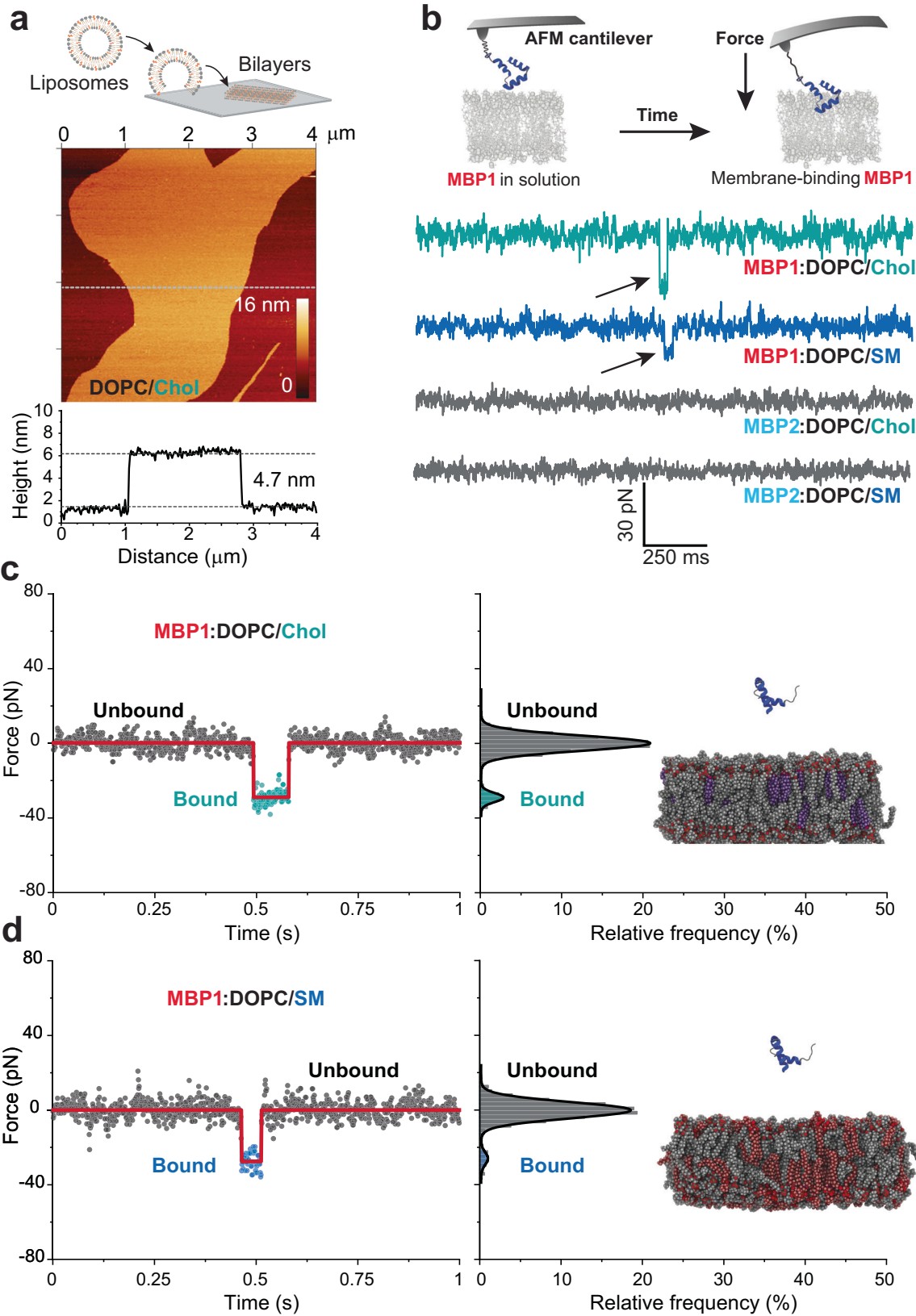

To investigate the atomistic details of the MBP interaction with the membrane, we mirrored our experiment in silico by performing microsecond-long MD simulations using QwikMD[45] and NAMD3[46]. The structures predicted by AlphaFold2 had pLDDT scores of 81.4 for MBP1 and 78.4 for MBP2, both of which fall within the 'confident' range according to AlphaFold[47]. Notably, the alpha-helix regions exhibited even higher scores, exceeding 90, a threshold considered indicative of 'very high confidence' (Supplementary Fig. 5). MBP1 and MBP2 structures were equilibrated using the MD protocol and inserted into the same two membrane environments as used previously. Unbiased MD simulations were performed for a duration of 5 μs for each system.

**Fig. 2 | Monitoring SARS-CoV-2 MBP binding to lipid bilayers using AFM-based SMFS. a** Formation of lipid bilayers on mica support (top). Lipid vesicles were adsorbed onto freshly cleaved mica surface, forming lipid bilayers. AFM was used to obtain the height topography of DOPC/Chol bilayers, and the height cross-section profile was extracted from the corresponding height image. **b** Schematic illustrating the binding of MBP to the lipid bilayers monitored by AFM-based SMFS. Representative force vs time (FT) curves showing the binding of MBP1 into the lipid bilayers (indicated by black arrowhead). MBP2 did not exhibit binding events with

the lipid bilayers. Representative FT curves of MBP1 binding to the DOPC/Chol membrane (**c**) or DOPC/SM membrane (**d**). Two distinct states (unbound: grey; bound: cyan/blue) were detected in the FT curves which were identified using the STaSI algorithm (red line) and depicted as histograms fitted using a multipeak Gaussian fit. Data points are color-coded based on the assignment by the STaSI model: gray dots represent the unbound state and cyan/blue dots the bound state. Source data for panels a-d are provided as a Source Data file. Data are representative of at least three independent replicates.

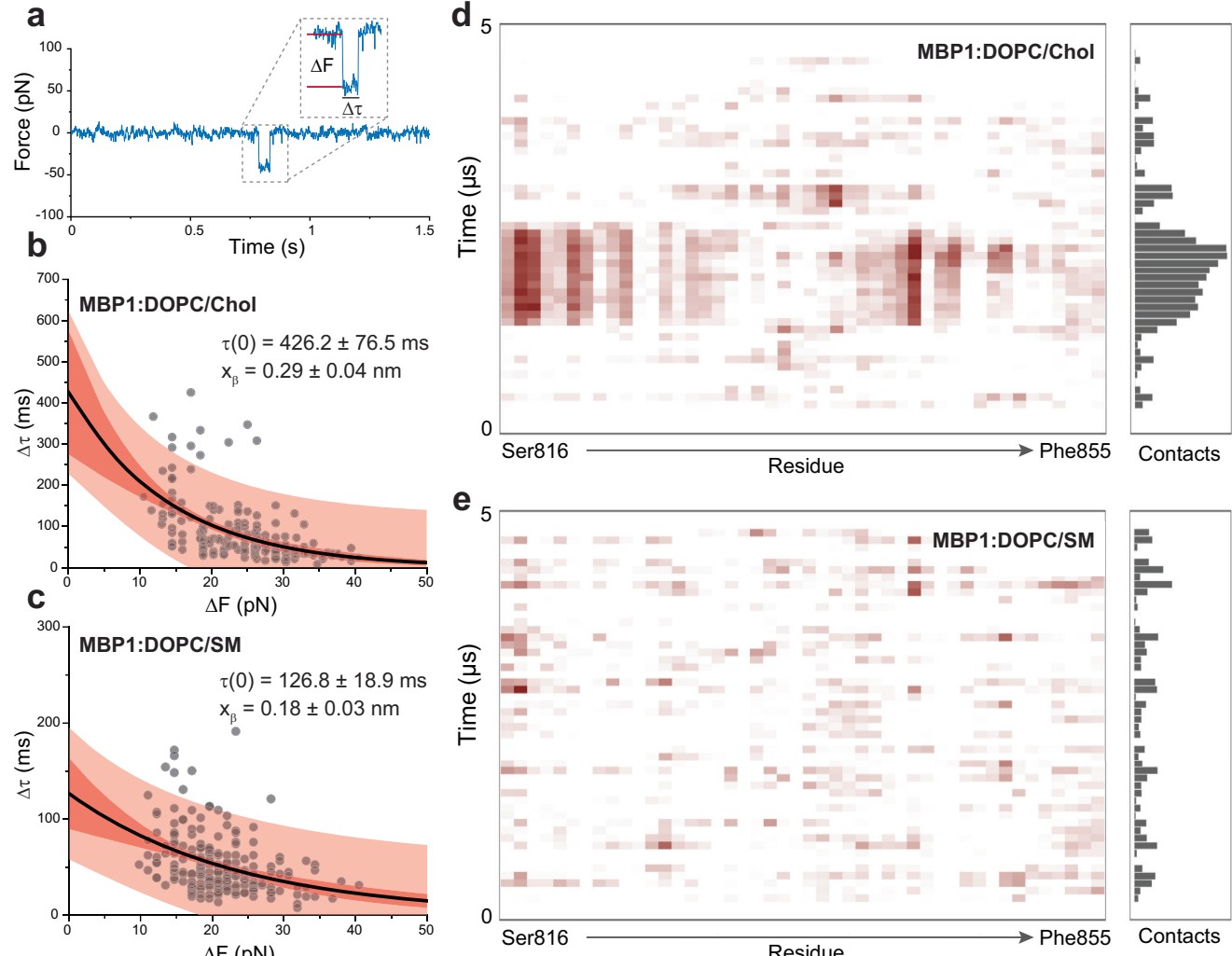

**Fig. 3 | Towards the kinetics of SARS-CoV-2 MBP binding to lipid bilayers. a** Representative force versus time curve, highlighting how the binding force (ΔF) and lifetime (Δτ) of the binding events were extracted. Graph showing the distribution of the data points extracted from (**b**) DOPC/Chol lipid bilayer (n = 180 from at least three independent replicates) or from (**c**) DOPC/SM lipid bilayer (n = 206 from at least three independent replicates) where n represents the number of insertion events quantified). The Bell model was then employed to fit the data

(black lines), providing an average $x_\beta$ and $\tau(0)$ values for the binding complexes. The darker shaded areas indicate 95% confidence intervals and lighter shaded areas represent 95% prediction intervals. Average contact between the amino acids of MBP1 and the DOPC/Chol membrane (**d**) or DOPC/SM (**e**) over time. Source data for panels a-c are provided as a Source Data file. Data are representative of at least three independent replicates.

The simulations revealed that MBP1 did not exhibit a stable secondary structure at the membrane interface, although a preference for maintaining an alpha-helix in most of the peptide was evident. Surprisingly, in our simulations, we did not observe the peptide fully intercalating into the lipid. In contrast, some other viruses, like flaviviruses, exhibit a behavior where the membrane-binding peptide integrates into the membrane within a few nanoseconds of initial contact. Under our simulated conditions, a simulation time of 5 μs should have been adequate to detect such insertion[48]. To

better understand the results, we calculated the average contact between the amino acids of the MBPs and the lipid membrane. We defined a contact as any amino acid atom within 4.5 Å of any membrane atom. The calculations performed every ns were averaged over 100 ns windows. The contact map reveals that MBP1 has a stronger preference for interacting with the cholesterol-enriched membrane as compared to the sphingomyelin membrane (Fig. 3d, e). A similar result was observed for MBP2 (Supplementary Fig. 6), wherein the interactions with the cholesterol-rich membrane appear more

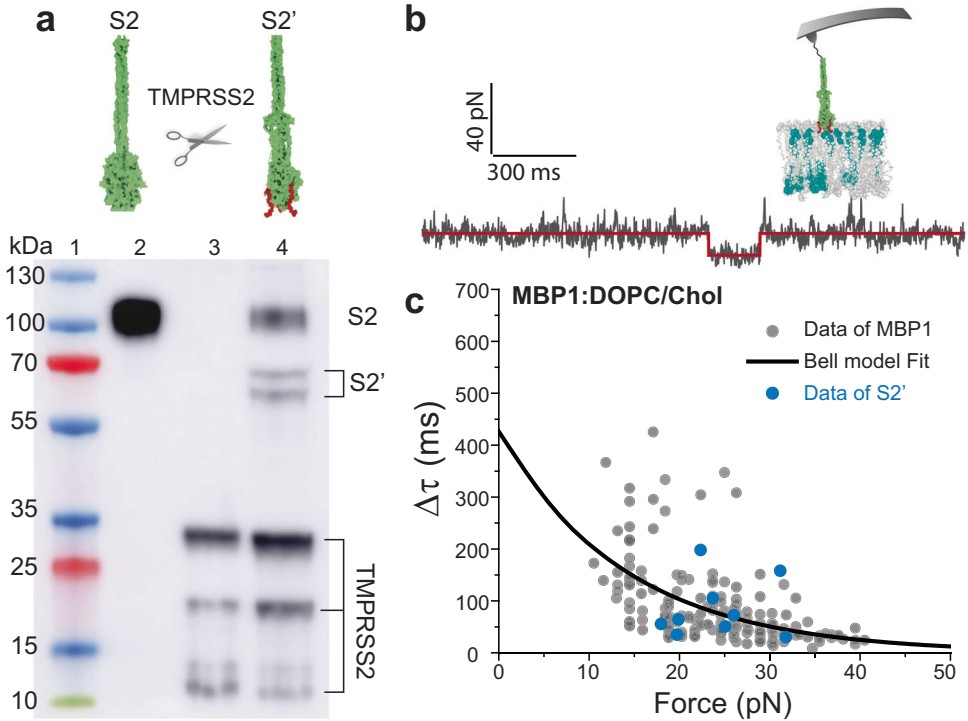

**Fig. 4 | Probing the binding of SARS-CoV-2 MBP at the S2 subunit level.**
**a** Western blot analysis of S2 subunit treated with TMPRSS2 under reducing conditions. The cleaved S2 subunit (S2') was detected using an antibody against the His-tag. Lane 1: Ladder, Lane 2: S2 subunit, Lane 3: TMPRSS2, and Lane 4: S2 subunit treated with TMPRSS2. The uncropped scan is provided as Source Data file. Image representative from 3 independent replicates. **b** Representative FT curves depicting the binding of S2' subunit to the DOPC/Chol bilayer. Raw data is plotted as a black line, and the binding states are identified using the STaSI algorithm (red line). **c** Plot of force-dependent lifetime obtained for S2' subunit (blue dots, $n = 9$ from at least three independent replicates) and overlaid on the data obtained for the MBP (gray dots, from Fig. 3b). Source data for panels b,c are provided as a Source Data file.

prevalent. The convergence of our experimental and computational findings highlights the pivotal role of MBP priming and membrane composition, particularly the presence of cholesterol, in modulating the binding affinity to SARS-CoV-2 MBP.

## Monitoring MBP binding at the S2 subunit level

Upon binding of host receptors to the S1 subunit and subsequent processing of the S2-N terminus site by the TMPRSS2 protease, the MBP of SARS-CoV-2 becomes exposed and is expected to bind to the host cell membrane, initiating the binding process between the virus and the host cell[19]. To test the physiological relevance of our results obtained at the short peptide level, we next performed a similar experiment at the S2 level. S2 subunit was attached to the AFM tip using the NHS/EDC covalent ligation strategy. The grafted sequence encompasses a significant portion of the N-terminal region, which lies upstream of the S2 priming site cleaved by TMPRSS2, thus maintaining the S2 subunit in its pre-fusion conformation. In line with the previous experiments, individual FT curves were recorded on a DOPC/Chol membrane, but no binding events were observed in the 453 force traces collected during this process. Following this, an enzyme cleavage experiment was conducted in situ following a previously established protocol[49]. The S2 subunit on the tip was enzymatically processed and cleaved by the TMPRSS2 enzyme, generating the S2' subunit (Supplementary Fig. 3b). This reaction was validated in parallel by a western blot assay (Fig. 4a).

After cleavage by TMPRSS2, individual FT curves demonstrated specific binding of the S2' domain to the DOPC/Chol membrane (Fig. 4b), with 9 binding events observed out of 339 FT curves recorded. These findings provide conclusive evidence that the TMPRSS2-processed S2 subunit binds to the host membrane via its MBP domain. Additionally, we extracted specific binding lifetimes and corresponding forces from the FT curves obtained using S2'-functionalized AFM tips. Notably, these binding parameters were in excellent agreement with previously acquired data for the SARS-CoV-2 MBP (Fig. 4c).

The experiments were conducted under conditions that mimic the viral membrane binding process, during which the S2 subunit remains in its pre-fusion state. This is supported by several factors: (i) the S2 subunit retains its N-terminal region above the S2 priming site, maintaining its pre-fusion conformation until activation by TMPRSS2; (ii) the AFM experiments were performed rapidly after TMPRSS2 activation, allowing for immediate observation of S2 priming; (iii) the data obtained align well with in vitro results for the MBP1 peptide, which lacks the fusion peptide; and (iv) the S2 subunit grafted onto the AFM tip lacks the viral membrane necessary for the transition from its pre-fusion to post-fusion states. Overall, these results confirm that the exposed MBP on the S2' subunit effectively binds to the host membrane and highlight the role of cholesterol in the membrane environment, which synergistically enhances the binding process.

## Role of the internal disulfide bond in altering the binding properties of MBP to lipid bilayer

The SARS-CoV-2 MBP has a conserved internal disulfide bond between Cys840 and Cys851 (Fig. 5a), which appears to play an important role in full MBP activity[20-22,50]. To better understand the role of this disulfide bridge in host membrane binding activity, we performed this analysis in the presence or absence of the disulfide bond in MBP1 (see Methods, section 'Functionalization of AFM tips and disulfide bond formation in peptide'). Analysis performed by mass spectrometry shows that the disulfide bridge was successfully reconstructed after treatment with dimethyl sulfoxide (DMSO) and disrupted after reducing tris(2-carboxyethyl) phosphine

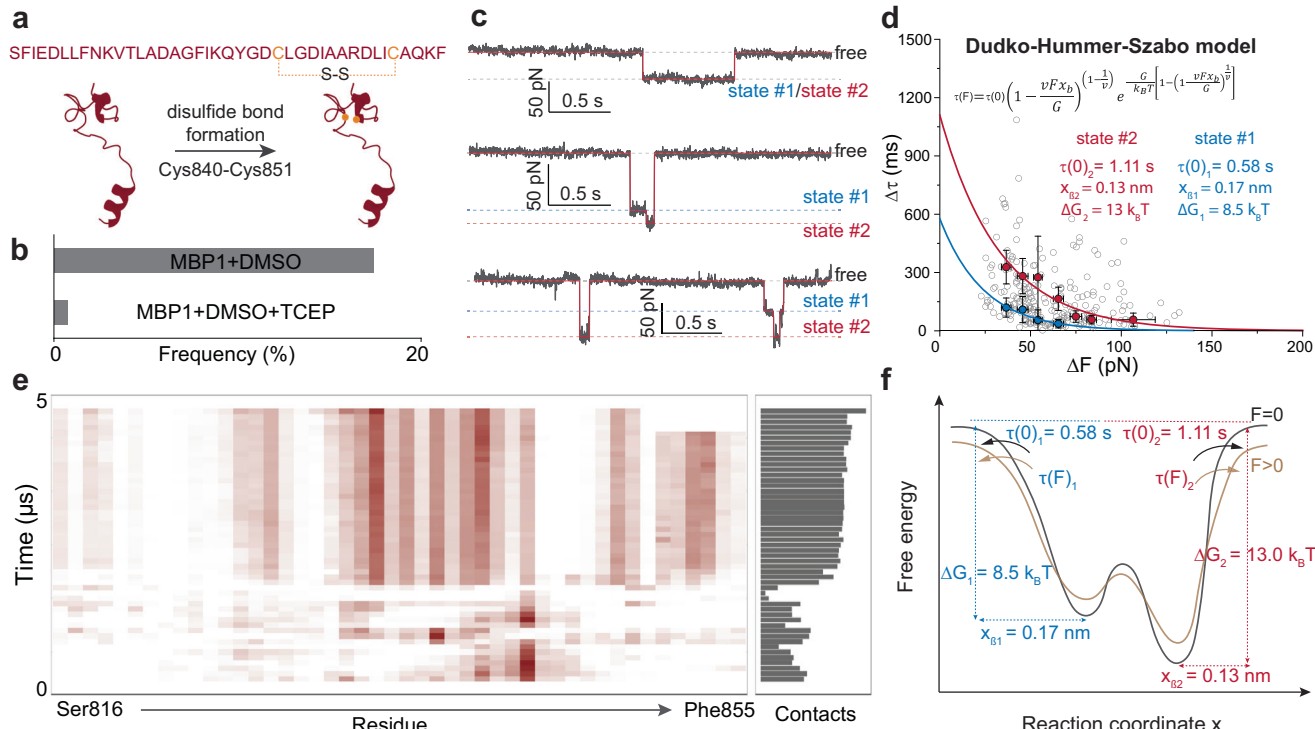

**Fig. 5 | Internal disulfide bond promotes MBP binding to the lipid bilayer. a** A disulfide bond is present in the MBP between Cys840 and Cys851. **b** The binding frequency of MBP1 after treatment with DMSO ($n = 237/1361$) or DMSO and subsequent reductive treatment with TCEP ($n = 2/267$). **c** curves with specific interaction events recorded after oxidation with DMSO. Raw data (gray line) are processed using the STaSI algorithm (red line) to identify different states in the force trace. **d** The bonds lifetime ($\Delta\tau$) of the MBP1 (treated with DMSO)-DOPC/Chol membrane bonds (gray dots) plotted against the binding force ($\Delta F$) (gray dots, $n = 237$ data points from at least three independent replicates). The colored circles show the average lifetime of the bonds determined for the different binding forces (Supplementary Fig. 8). The lifetime of the bonds is classified with the average values for single (blue) and double (red) peptide-membrane bonds produced simultaneously.

The error bars show the standard deviation (s.d.). The solid line shows the fit of the average bond lifetimes based on the DHS model. **e** MD simulations of MBP1 with intact disulfide bridge in interacting with the DOPC membrane with 30% Chol revealed an increase in the number of contacts over time as well as a higher number of residues all spread over the length of the MBP. **f** The free-energy landscape of the MBP binding to the membrane under a constant force. The kinetics and thermodynamics analyses of the bond lifetimes in the MBP1 peptide (treated with DMSO)-DOPC/Chol membrane interaction reveal specific binding comprising of two distinct states in its binding free-energy landscape, which can be explained by a conformational change facilitated by the presence of the disulfide bridge. Source data for **a**–**d** are provided as a Source Data file.

hydrochloride (TCEP) treatment (Supplementary Fig. 7). We further compared peptide binding in the absence or presence of this disulfide bridge at the single molecule level (Fig. 5a, see Methods, section 'AFM-based single-molecule force spectroscopy'). We observed specific interactions in about 17.5% of cases ($n = 237/1361$ for MBP1-DOPC/Chol) under conditions favoring the formation of the disulfide bridge, whereas the interactions were virtually abolished after reductive treatment with TCEP (Fig. 5b). Conversely, the specific strong interaction was not observed on the DOPC/SM lipid bilayer. These results strongly suggest that the main interaction events were due to both the SARS-CoV-2 MBP and the DOPC/Chol membrane.

To probe the dynamics of MBP binding to the membrane in the presence of an internal disulfide bond, we analyze the binding dynamics from the recorded FT curves. As depicted in the FT curves (Fig. 5c), following the initial binding event, a subset of the curves displayed an additional step towards higher forces, suggesting the existence of a distinct population or state. The STaSI algorithm effectively distinguishes between these two populations. The collected data reveals that lifetimes range from 15 to 570 ms for forces spanning 25 to 130 pN (Fig. 5d, and Supplementary Fig. 8a). This second population of data might indicate either a shift in binding conformation towards enhanced stability or the binding of a second MBP bound to the tip interacting also with the lipid membrane.

To discriminate between these two hypotheses, we analyzed the force distribution as a function of lifetimes extracted from the FT

curves. (Fig. 5d, and Supplementary Fig. 8a). The lifetime distributions were binned across seven discrete force ranges (Supplementary Fig. 9, #1 to #7). Notably, the lifetime distributions in the four lower force regions (#1 to #4, F < 70 pN) exhibited a distinctive bimodal Gaussian distribution (Supplementary Fig. 9), with the frequency of the second peak significantly less than the first one. The presence of a second population even for the lower force suggests the hypothesis of a conformational change induced by a change in the oxidation state of the internal cysteine residues.

According to the Bell model[44], the lifetime of a bond exponentially decreases as the force increases. By accounting for the complex nature of energy landscapes, the Dudko-Hummer-Szabo (DHS) model provides a more nuanced and accurate description of how molecular bonds behave under external force by considering the multi-dimensionality and shape of the energy barriers. Applying the Bell model, we determined the separation distance from the unbound to the bound state for the first population as $x_{\beta1} = 0.17$ nm and the lifetime ($\tau(0)_1 = 0.58$ s) (Supplementary Fig. 8b). Analysis of the second population revealed higher stability, characterized by a lower free-energy valley ($x_{\beta2} = 0.15$ nm) and a twofold higher lifetime ($\tau(0)_2 = 1.28$ s). To evaluate the binding free-energy differences between both populations, we next analyzed our data with the DHS model[51]. This model allowed us to simultaneously determine the activation energy ($\Delta G$) and kinetic parameters ($x_\beta$ and $\tau(0)$) by examining the lifetime-force relationship (Fig. 5d). We obtained values of

$x_{\beta 1} = 0.17$ nm and $\tau(0)_1 = 0.58$ s for the first population, and $x_{\beta 2} = 0.13$ nm and $\tau(0)_2 = 1.11$ s for the second population. These results were in excellent agreement with the values obtained using the Bell model. Notably, the binding free-energy for the second population ($\Delta G_2 = 13$ $k_B T$) is significantly higher than for the first population ($\Delta G_1 = 8.5$ $k_B T$), indicating a stronger stabilization of MBP. However, the fact that the energy of the second population is significantly lower than two times the energy of the first population again favors a conformational change to a more stable bound state rather than two peptides interacting in parallel.

To further test this possible second conformational state, we also investigated the atomistic details of the MBP with the disulfide bridge through MD simulations. The initial structure was constructed by modifying the system to form the disulfide bridge and performed the same MD protocol as the above systems. Our results revealed, upon encountering the membrane surface, the region between the two cysteines established strong contact and remained so for over 1 μs. This first contact nucleates around the Arg846 residue which matches also what we observed previously (Fig. 3d). A subsequent change in the peptide structure is then observed, slightly reducing this interaction for a short time, but then the entire peptide established closer contact with the membrane surface, which persisted for the remainder of the 5 μs-long simulations (Fig. 5e). To further validate the critical role of the Arg846 residue, we synthesized a mutant version of MBP1 in which Arg846 was replaced by Gly (MBP1 mutant). Using AlphaFold2, we confirmed that both the wild-type MBP1 and the MBP1 mutant peptides exhibit similar overall conformations, particularly at the site of the mutation. We then conducted AFM-based SMFS experiments using this point mutant peptide. Strikingly, the results showed that the MBP1 mutant failed to bind to the DOPC/Chol membrane under the tested conditions (Supplementary Figs. 4, 5), pointing out the critical role played by a single Arg846 residue. Additionally, some lipid tails began to flip and interact with the MBP1, indicating a stronger peptide binding to the membrane. Surprisingly, we never observed the MBP1 completely across the lipids or aligning along the Z-axis. Also, although the membrane was rich in cholesterol, we did not observe any major direct contact between the peptide and the cholesterol. This configuration is consistent with the recent structural findings, which show that the sequence (aa 816-856) of S protein has a half-contact with the membrane and may also contribute to anchoring the post-fusion S2 structure in the membrane[42]. These findings corroborate our experimental observation and are suggestive of a more second stable state.

In vitro and in silico data converge on a well-defined progressive atomistic mechanism, in which initial contact is established between the short loop formed between Cys residues 840-851. This contact is followed by a conformational change allowing increased contact with residues further upstream and downstream. The binding complex between the MBP and the DOPC/Chol membrane can be described as a binding-free energy landscape with two valleys, the second conformational state exhibiting an extended lifetime and heightened binding free-energy barrier (Fig. 5f). We also propose that the disulfide bridge in addition to imparting enhanced mechanical stability to the MBP-membrane complex also helps in maintaining spatial proximity between the membrane and the peptide. This proximity promotes a concerted process by fostering the conformational change and subsequent attachment of new residues, contributing to the overall stepwise binding process.

### Cholesterol depletion inhibits SARS-CoV-2 cell infection

At the single-molecule level, we observed a strong influence of cholesterol on the binding of SARS-CoV-2 MBP. Cholesterol within the viral envelope can modulate the fusion process by affecting the curvature and fluidity of the membrane, which are essential for the fusion between the virus and host cell[52,53]. This led us to investigate whether the cholesterol level in the host cell membrane have a bearing on the

susceptibility to SARS-CoV-2 infection. Given its crucial role in maintaining membrane integrity, it also facilitates the entry of enveloped viruses and hence local or global variations in cholesterol levels could potentially alter the efficiency of viral attachment and fusion with the host cell membrane. This suggests that higher or lower cholesterol content in cell membranes might modulate how effectively the virus infects the host cells, thereby impacting the overall infection process. To investigate this, we conducted virus infection assays using pseudotyped viruses carrying S proteins. A549 cells expressing ACE2 and TMPRSS2 were incubated with propagation-incompetent G-deleted vesicular stomatitis virus (VSV) trans-complemented with the SARS-CoV-2 spike protein and encoding a GFP reporter protein (VSV-SARS-CoV-2). A549 cells were infected with the VSV-SARS-CoV-2. The infectivity was assessed by measuring the GFP fluorescence 24 hours post-infection. To assess the role of cholesterol, A549 cells were treated with varying concentrations of Methyl-β-cyclodextrin (MβCD), a commonly used agent for cholesterol depletion from the plasma membrane[54]. The results revealed a concentration-dependent inhibition of VSV-SARS-CoV-2 infection by MβCD (Fig. 6a, b), consistent with previous findings[30]. To further confirm the cholesterol-dependence of VSV-SARS-CoV-2 infection, A549-ACE2-TMPRSS2 cells were treated with Methyl-β-cyclodextrin (MβCD) to deplete cholesterol and subsequently supplemented with exogenous cholesterol. The results showed that SARS-CoV-2 infection increased more than twofold following cholesterol repletion in MβCD-treated cells. This indicates that cholesterol depletion significantly impairs viral entry, which can be effectively restored by adding exogenous cholesterol. However, in cells that were not treated with MβCD, the addition of exogenous cholesterol resulted in only a slight increase in infection (Fig. 6c, d). These infectivity assays demonstrate that cholesterol is essential for efficient viral binding and entry, likely by increasing the binding affinity of viral membrane-binding peptide to host membrane, consistent with our in vitro findings that the MBP within the S2 domain interacts with cholesterol after TMPRSS2 priming to facilitate membrane attachment. Cholesterol levels may also influence subsequent steps such as membrane fusion.

### Discussion

In the context of SARS-CoV-2 infection, the binding of the MBP to the host membrane emerges as a pivotal event crucial for viral entry and subsequent infection. This interaction facilitates the fusion of viral and cellular membranes, enabling the release of the viral genome into the host cell cytoplasm, ultimately leading to viral replication and spread. Chol stands out among lipids as a critical factor influencing the membrane fusion of SARS-CoV-2[28–30]. It initially reduces the energetic demand for interactions between the lipid bilayers of both the virus and the cell due to its capability to stabilize hydrophobic interactions. However, so far, the molecular details describing the MBP binding kinetics and thermodynamics to host membrane is poorly understood experimentally, although it represents a key target for therapeutic intervention.

In our study, we utilized a combination of in vitro experiments and in silico simulations to investigate the role of Chol in enhancing the interaction between the MBP of SARS-CoV-2 and the host cell membrane. Through BLI, we observed that the binding affinity of SARS-CoV-2 MBP to DOPC/Chol membranes is five times greater than that to DOPC/SM membranes. This finding was corroborated by AFM-SMFS experiments and MD simulations, where we directly observed specific MBP binding to Chol-enriched membranes at the individual peptide level. Our kinetic analysis demonstrated that MBP binding to DOPC/Chol membranes exhibits a significantly prolonged lifetime compared to the DOPC/SM membrane control, indicating the crucial role of Chol in stabilizing the bound complex. Notably, we observed that maintaining the integrity of the disulfide bridge within the MBP enhances the mechanical stability of its binding. This stability is

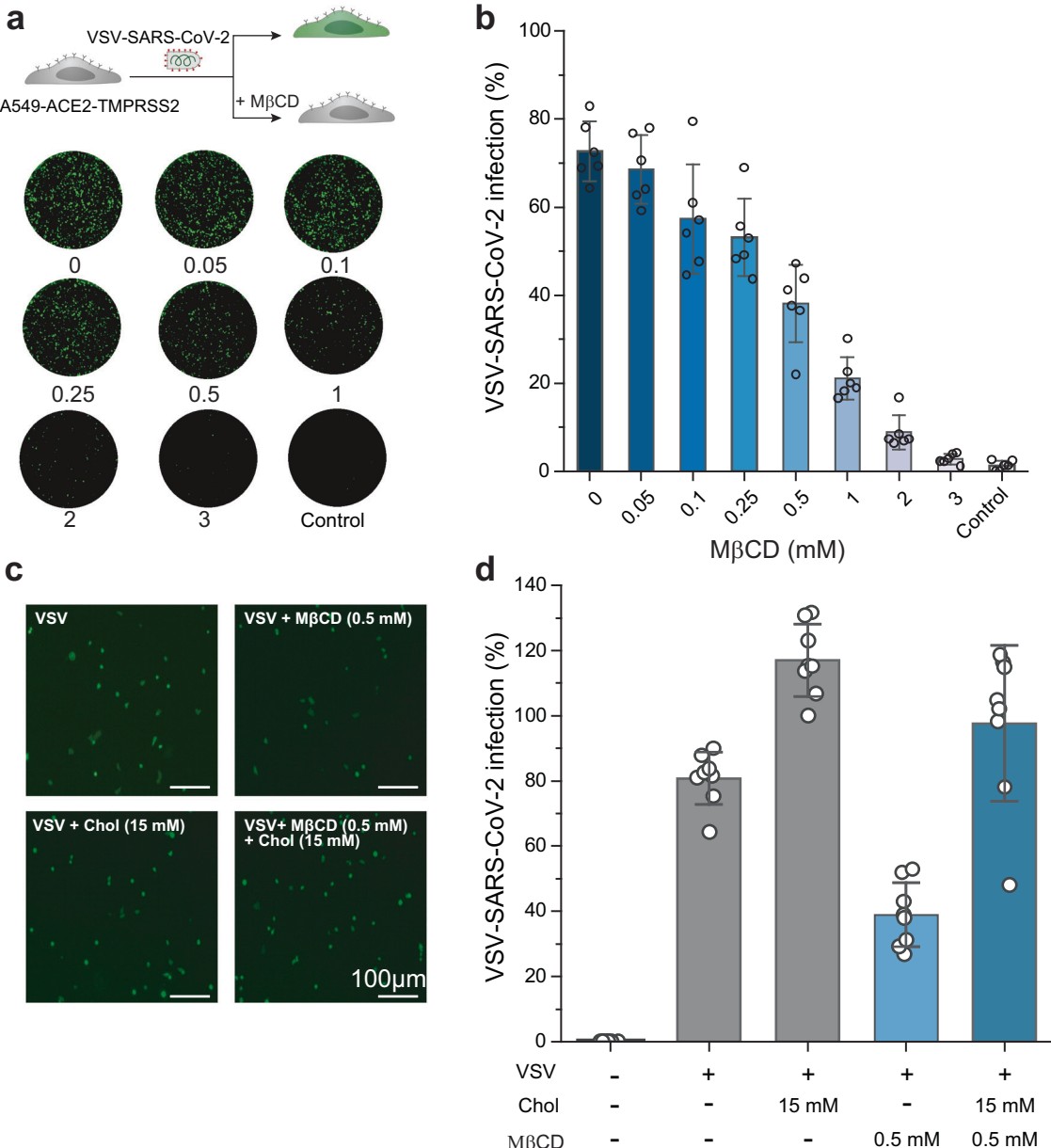

**Fig. 6 | Chol depletion using MβCD inhibits SARS-CoV-2 pseudovirus entry into host cells. a** Schematic representation of the SARS-CoV-2 pseudovirus infectivity assay using A549 cells expressing ACE2 and TMPRSS2 (top). Fluorescent images of infected A549-ACE2-TMPRSS2 cells in the presence of different concentrations of MβCD (0, 0.05, 0.1, 0.25, 0.5, 1, 2, 3 mM), and control without virus (bottom). **b** Mean infectivity was measured at various concentrations of MβCD. The virus infection is expressed as percentage and normalized by the control experiment. Each black dot represents infectivity from a well. **c**, **d** The exogenous-cholesterol supplement assay. The A549-ACE2-TMPRSS2 cells were treated with MβCD (0.5 mM) for 1 h, and then supplied with exogenous Chol (15 mM) for 1 h. Each black dot represents infectivity from a well. The column indicates the mean value, and the whiskers indicate the standard deviation (s.d.) of the mean. Source data for panels **b**, **d** are provided as a Source Data file. Data are representative of three independent replicates.

---

evidenced by an initial binding to the Chol-enriched membrane, followed by a conformational change that leads to a more stable secondary state. Previous reports have suggested that MBP attaches rapidly to cellular membranes without fully crossing the lipid bilayer[21,22,55], consistent with our findings. In the present work, we experimentally quantify the binding kinetics and thermodynamics of MBP to the membrane and demonstrated the role of the internal conserved disulfide bridge in the MBP domain in establishing the strong binding necessary for SARS-CoV-2 membrane fusion. Furthermore, the single point mutation (Arg846) in MBP is sufficient to prevent its attachment to the membrane.

Additionally, our findings reveal that the infection mediated by SARS-CoV-2 pseudotyped particles is impeded following Chol removal

from the host membrane. Collectively, these findings suggest that Chol might have a crucial involvement in SARS-CoV-2 infection and could be a viable target for developing effective therapies against the virus. This aligns well with prior studies demonstrating the antiviral effects of cholesterol-25-hydroxylase[56,57] and statins[58] against SARS-CoV-2, observed both in vitro and in vivo.

Beyond the MBP-lipid interaction analyzed in this study, the spike protein's S2 subunit plays an equally crucial role in mediating the virus-host membrane fusion process. Following cleavage by TMPRSS2, the S2 subunit undergoes substantial conformational changes, transitioning from its prefusion to postfusion states, as described in recent high-resolution cryo-EM studies[59]. These changes lead to the formation of the six-helical bundle, which is essential for bringing the viral and

host membranes into close proximity and enabling fusion. TMPRSS2-mediated cleavage has been shown to be vital for the activation of the S2 subunit and its role in fusion[60,61]. While our work has focused on the MBP-lipid membrane interaction, the S2 subunit's dynamics post-cleavage also warrant further investigation, as they are integral to the fusion process. Future studies could explore both in vitro and in silico dynamics of the S2 post-cleavage transition to provide a more holistic view of membrane fusion.

The MBP emerges as a critical component in the context of SARS-CoV-2. In contrast to the extensive accumulation of mutations observed in the SARS-CoV-2 RBD, the S2 fusion subunit has maintained significant conservation across variants[62]. As such, it could be an ideal target for the design of cross-protective fusion inhibitors and vaccine candidates against emerging coronaviruses.

In conclusion, this study illuminates the role of Chol in enhancing the binding of SARS-CoV-2 MBP to the lipid membrane, with this process notably reliant on MBP priming via TMPRSS2. The MBP loop crosslinked by the disulfide bond emerges as a key factor stabilizing the MBP-membrane complex. Additionally, the Chol content within the host membrane significantly influences SARS-CoV-2 infection. Our findings not only provide a mechanistic understanding of how Chol crucially boosts SARS-CoV-2 infection but also offer insights for devising broadly effective therapies capable of addressing emerging variants and potential future coronavirus outbreaks.

## Methods

### Reagents and Membrane binding peptides

Brain Sphingomyelin (SM; Avanti Polar Lipids); Cholesterol (Chol; Sigma); 1,2-dioleoyl-sn-glycero-3-phosphocholine (DOPC; Avanti Polar Lipids); DPBS (Fisher); Sortase A (BPS Bioscience, San Diego, CA, USA); NHS-PEG$_{24}$-Ph-aldehyde spacer (Broadpharm, San Diego, CA, USA); poly-Glycine linker (Genscript, NJ, USA), S2 subunit His-tag (10594-CV, BioTechne, Germany), TMPRSS2 (Biozol,CSB-YP023924HU). Unless stated otherwise, all chemical reagents were purchased from Sigma (Missouri, USA). The membrane binding peptides (MBP1 and MBP2) of SARS-CoV-2 with a purity of >95% were purchased commercially from GenScript. Peptide purity and structural integrity were validated using HPLC and MS. The following peptides were investigated:

MBP1: [816]SFIEDLLFNKVTLADAGFIKQYGDCLGDIAARDLICAQKF[855]LPETGG)

MBP2: [806]ILPDPSKPSKRSFIEDLLFNKVTLADAGFIKQYGDCLGDIAARDLICAQKF[855]LPETGG

MBP1 mutant: [816]SFIEDLLFNKVTLADAGFIKQYGDCLGDIAA**G**DLICAQKF[855]LPETGG

MPP: [866]TDEMIAQYTSALLAGTITSGWTFGAGAALQIPFAMQMAYRFNGI[910]GGGGGLPETGG

The peptides were dissolved in DPBS and stored at −20 °C in 20 µl small stock aliquots.

### Preparation of lipid vesicles

DOPC, brain SM, and Chol were dissolved in chloroform to give a final lipid concentration of 5 mM. Aliquots of DOPC and SM or Chol solutions were mixed in DOPC/SM and DOPC/Chol in molar ratios (70/30) and poured into a glass vial. Then, the resultant solutions were dried under a nitrogen flow and kept for 20 min in a vacuum desiccator to ensure complete removal of chloroform. Multilamellar vesicles were obtained by hydration with the Buffer A (20 mM HEPES and 50 mM NaCl at pH 7.4) solution to give a final lipid concentration of 1 mM, as previously described[33]. To obtain the lipid vesicles, the mixed solution was extruded through Whatman Nuclepore Hydrophilic Membrane with a 100 nm pore size using an Avanti Mini-Extruder (Avanti Polar Lipids, Alabama, USA).

### Supported lipid bilayers for AFM-based SMFS experiments

Supported lipid bilayers were prepared on circular mica surfaces for AFM-based SMFS experiments as described previously[33]. In brief, the lipid vesicles were dropped on the freshly cleaved mica surfaces cemented onto Teflon discs with epoxy-based mounting glue. A metal puck was used as a supporting substrate for the Teflon discs and were magnetically mounted on the AFM stage. The deposited small lipid vesicles on the surface were incubated at 60 °C for 1 h to obtain the lipid-supported bilayers. The samples were rinsed with Buffer B (20 mM HEPES, 50 mM NaCl, and 2 mM CaCl$_2$ at pH 7.4) maintained at 60 °C and then slowly cooled to room temperature (R.T).

### Functionalization of AFM tips and disulfide bond formation in peptide

To functionalize peptide/protein on AFM tips, MBPs and S2 subunits were grafted onto AFM tips apex (AC40, Bruker), using the following protocol: the AFM tips were washed in chloroform (3 × 10 min), cleaned in a UV-ozone (UV-O) cleaner (Jelight, California, USA). The tips were further incubated for 2 h in an argon-filled desiccator with 30 µL APTES and 10 µL triethylamine. To cure the APTES coating, the tips were maintained under argon for 2 days. To graft the respective peptide or protein on the AFM tip, a heterobifunctional NHS-PEG$_{24}$-Ph-aldehyde was grafted on the amino-functionalized tips. NHS-PEG$_{24}$-Ph-aldehyde (3.3 mg) was dissolved in chloroform (0.5 mL) and tri-methylamine (30 µL). The cantilevers were immersed in the solution for 2 hours and washed 3 times with chloroform and dried with nitrogen. Subsequently, the cantilevers were immersed in the 100 µL Gly$_{10}$Lys peptide (1 mM) or S2 subunit (300 nM) (Wuhan-1, Accession#: YP_009724390, collected on 24 December 2019, R&D Systems) solution with 2 µL fresh NaCNBH$_3$ at 4 °C for 1 h. Then, 5 µl of 1 M ethanolamine solution (pH 8.0, 10 min, R.T) was added to the solution to quench the unreacted aldehyde groups and washed 5 times with DPBS buffer. The S2 subunit-functionalized AFM tips were directly used for binding assays. To obtain MBPs-functionalized AFM tips, the Gly$_{10}$Lys peptide-modified tips was then incubated with 50 µl of a 10 µM MBPs and 50 µl of a 20 µM Sortase A solution for 4 h at 37 °C. MBPs-tips were rinsed with HEPES buffer and stored at 4 °C and used in AFM experiments on the same day.

An efficient method for disulfide bond formation in peptide is described[63]. The AFM tip of the functionalized MBP1 was rinsed by HEPES buffer. This tip was reacted with 20% dimethyl sulfoxide (DMSO) at pH 6 in room temperature for 4 h. This tip was then rinsed with HEPES buffer and stored in 4 °C until used in AFM experiments.

### AFM-based single-molecule force spectroscopy

A Bioscope Resolve AFM (Bruker) with a 100 µm piezoelectric scanner was used to image the lipid bilayers with PeakForce QNM mode (Nanoscope software v9.2, Bruker, Santa Barbara, USA). Using the bare or functionalized cantilevers, the membrane patches were imaged with an imaging force of 250 pN, 100 nm ramp size, and 256 × 256 pixels and scan size of 10 × 10 µm. After finding membrane patches, the tips were moved closer to the membrane about 100 pN in the height clamp mode. The AFM tip was then retracted 6 nm away from the surface. Next, the cantilever was kept at a constant height for 10 s while recording the cantilever deflection to collect FT curves. After that time, the cantilever was retracted by 50 nm and the measurement were repeated. Cantilever spring constants were calibrated using the thermal tune method and deflection sensitivity was measured by ramping on hard bare mica surfaces. All experiments were performed at room temperature and repeated at least five times.

### Analysis of FT curves and extraction of kinetic parameters

After excluding non-specific binding events by Nanoscope analysis 2.0, FT curves with the binding events were run to correct the baseline using an asymmetric least-squares baseline algorithm in Origin 2021 software. To analyze specific binding events, we adapted the Step Transition and State Identification (STaSI) algorithm[43], originally

designed for discrete single-molecule data, to suit our dataset. The STaSI algorithm identifies step transitions using Student's t-test and subsequently groups segments through hierarchical clustering. The optimal number of states is determined by balancing model complexity and goodness of fit, applying the principle of minimum description length to achieve the simplest model with the least fitting error. The algorithm was rigorously validated using simulated traces featuring two states.

Kinetic and thermodynamic parameters of the binding events were estimated by applying the Bell model[39,44] (Fig. 3) through non-linear curve fitting in Origin 2021. Data from force-clamp experiments were analyzed using the Bell model equation:

$$\tau(F) = \tau(0) * e^{-\frac{F x_\beta}{k_B T}} \qquad (1)$$

where $\tau(F)$ is the complex of MBP-membrane lifetime as a function of force, $\tau(0)$ is the bond lifetime in equilibrium, $x_\beta$ is the width of the energy valley, $k_B$ is the Boltzmann constant, and T the temperature.

To evaluate the free energy of activation energy ($\Delta G$), Dudko-Hummer-Szabo (DHS) is described[51]. The DHS formalism introduces an additional parameter to the BE model, allowing for smooth interpolation between different potential shapes by incorporating the apparent free energy of activation ($\Delta G$). This model was validated as a straightforward approach for determining force-dependent lifetimes $\tau(F)$ under constant force conditions:

$$\tau(F) = \tau(0)\left(1 - \frac{\upsilon F x_\beta}{\Delta G}\right)^{(1-\frac{1}{\upsilon})} e^{-\frac{\Delta G}{k_B T}\left[1 - \left(1 - \frac{\upsilon F x_\beta}{\Delta G}\right)^{\frac{1}{\upsilon}}\right]} \qquad (2)$$

where $\tau(F)$ is the complex of MBP-membrane lifetime as a function of force, $\tau(0)$ is the bond lifetime in equilibrium, $x_\beta$ is the width of the energy valley, $k_B$ is the Boltzmann constant, and T the temperature. The scaling factor specifies the nature of the underlying free-energy profile: a = 1 corresponds to the BE expression. This corresponds to a linear-cubic potential, which is similar to the behavior assumed in the classical Bell model but adjusted to account for the curvature of the barrier[51].

## Biolayer interferometry assay

Kinetic analysis was performed on Amine Reactive Second-Generation (AR2G) biosensors (Sartorius) on an Octet BLI (BLI, Octet, Sartorius, Göttingen, Germany). First, the sensors were activated with EDC (1-Ethyl-3 [3-dimethylaminopropyl] carbodiimide hydrochloride) and NHS (N-hydroxysulfosuccinimide) to generate highly reactive NHS esters. MBPs (10 μM) were immobilized on the activated sensor surface. After this covalent grafting step, the unreacted ester groups were quenched with 1 M ethanolamine solution (pH 8.5), following which the resultant sensors were made to react with a serial dilution of lipid vesicles (0.075, 0.125, 0.25, 0.5, and 1 mM in PBS) for 300 s. Subsequently, the biosensors were allowed to dissociate in a PBS for 300 s. All measurements were conducted at RT and 1000 rpm shaker speed. The data were analyzed using the Octet Data Analysis 11.0 software (Octet) using a 1:1 isotherm model to calculate respective dissociation constants.

## Molecular dynamics simulations

Membrane models for this study were constructed using CHARMM-GUI[64], employing its Membrane Builder plugin[65]. Two lipid compositions were utilized, both featuring 1,2-Dioleoyl-sn-glycero-3-phosphocholine (DOPC) as the primary component. One membrane included 30% cholesterol, while the other comprised 30% sphingomyelin. The membranes dimensions were set to 140 Å along both X and Y axes, while Z-axis size was set according to the water layer thickness of 22.5 Å both +Z and −Z directions. The system was solvated

in TIP3 water with 0.15 mol/L NaCl, which also neutralized the system charge. Each system contained approximately 300,000 atoms. Both membranes underwent molecular dynamics simulations for minimization and equilibration using NAMD 3[46]. The simulations began with 10,000 steps of minimization, followed by 10 ns of initial equilibration, during which the phosphate atoms of the membrane were restrained along the Z axis. This initial equilibration phase was succeeded by 10 ns of unrestrained MD simulations. All MD equilibrations were performed in the NPT ensemble with a 2.0 fs timestep.

Due to the lack of experimentally resolved structures, the peptides were modeled using AlphaFold2[47]. To simulate the presence and absence of a disulfide bridge between the cysteines, systems were prepared with and without the bond, using VMD's autopsfgen[66]. The reliability of the AlphaFold2 models was assessed using the Predicted Alignment Error (PAE) and the predicted Local Distance Difference Test (pLDDT) scores. The structures predicted by AlphaFold2 underwent minimization and equilibration protocols using NAMD 3[46]. Using QwikMD[45], the peptides were solvated in TIP3 water with 0.15 mol/L NaCl, following standard MD protocols implemented in QwikMD. The minimization process was carried out for 5000 steps. Minimization was followed by 1 ns of MD simulation with position restraints on the peptide's backbone atoms to pre-equilibrate the peptides in the solvent. Lastly, we conducted 10 ns of unrestrained MD simulations. All equilibration MD simulations were performed in the NPT ensemble with a timestep of 2.0 fs.

The structural files of the membranes and peptides were combined by positioning the peptides 40 Å away from the membrane's center in VMD[67]. Clashing water molecules were removed, and the ionization state was adjusted to maintain a 0.15 mol/L concentration of NaCl. Subsequently, we performed 10,000 steps of energy minimization, followed by MD simulations using a temperature ramping protocol, gradually increasing the system's temperature from 0 K to 310 K over 0.5 ns to pre-equilibrate the entire system. In total, five simulation systems were built.

All MD simulations were conducted using NAMD 3[46], utilizing the CHARMM36 force field[68] for an accurate representation of biomolecules. Simulations were executed in a GPU-resident mode on DGX-A100 nodes at Auburn University. For each system, production runs were carried out in the NPT ensemble for a total duration of 5 μs. In total, over 25 μs of MD simulations were performed in this study. The temperature was maintained at 310 K using Langevin dynamics with a damping coefficient of 5 ps⁻¹. The pressure was maintained at 1 bar using a Langevin piston, with a piston period of 100 fs and a decay time of 50 fs. A time step of 2 fs was employed for integrating the equations of motion. For non-bonded interactions, a cutoff distance of 12 Å was applied to short-range interactions, while long-range electrostatic forces were calculated every two steps using the Particle Mesh Ewald (PME) method with a grid spacing of 1 ˚A[69] Lennard-Jones potentials were computed within a cutoff distance of 12 Å, with a switching function starting at 9 Å. The SHAKE algorithm was utilized to constrain all bonds involving hydrogen atoms, enabling the use of a longer time step. The trajectories generated during the production phase were analyzed using VMD[67] and custom Tcl and Python scripts. Key properties such as root-mean-square deviation (RMSD), root-mean square fluctuation (RMSF), and hydrogen bonding patterns were calculated to assess the stability and dynamics of the protein-membrane interactions. Additionally, VMD's timeline tool was employed to calculate the prevalence of interactions between the MBP and the lipid membrane.

## TMPRSS2 cleavage of S2 and western blotting

To confirm the S2 subunit is cleaved by TMPRSS2 (Biozol), the 5 μl (1.5 μM in Tris buffer) purified S2 was incubated with 20 μl (5 μM) TMPRSS2 in 50 mM Tris buffer (0.2% Triton X-100, 50 mM NaCl, pH 7.2) overnight at 37 °C, which was analyzed by Western blot. The

cleaved S2 subunit was run in SDS-polyacrylamide gel electrophoresis and subsequently transferred onto a PVDF membrane (Biorad, California, USA). The membrane was blocked with 1% BSA in TBS solution with 0.1% Tween-20 for 1 h, incubated with 0.5 μg/ml of Rabbit anti-His-tag (Abcam, ab9108) overnight at 4 °C, followed by incubation with secondary antibody Goat anti-Rabbit Horseradish peroxidase (HRP) (Agilent). For chemiluminescence detection, the membrane is briefly washed with the mixture of substrate A and substrate B (Pierce ™ ECL Plus Western Blotting Substrate, Thermofisher) with a ratio of 40:1. Then, the membrane was analyzed by Amersham Imager 600 (Cytiva). The S2-functionalized AFM tips were treated with TMPRSS2 under the same conditions with cleavage of S2 in vitro. Mock treatment was conducted without enzymes.

### Mass spectroscopy

Peptides were diluted in water/acetonitrile (5%), 0.1% formic acid at a concentration of 5 ng/μl. The peptide mixture was separated by reverse phase chromatography on a NanoACQUITY UPLC MClass system (Waters) working with MassLynx V4.1 (Waters) software. Briefly, 5 μl of each were injected on a trap C18, 100 Å 5 μm, 180 μm x 20 mm column (Waters) and desalted using isocratic conditions with at a flow rate of 15 μl /min using a 99% formic acid and 1% (v/v) ACN buffer for 3 min. The peptide mixture was subjected to reverse phase chromatography on a C18, 100 Å 1.8 μm, 75 μm x 150 mm column (Waters) PepMap for 35 min at 35 °C at a flow rate of 300 nl/min using a two-part linear gradient from 1% (v/v) ACN, 0.1 % formic acid to 40 % (v/v) ACN, 0.1 % formic acid for 15 min and from 40% (v/v) ACN, 0.1 % formic acid to 85 % (v/v)) ACN, 0.1 % formic acid for 10 min. The column was re-equilibrated at initial conditions after washing 30 min at 85% (v/v)) ACN, 0.1 % formic acid at a flow rate of 300 nl/min. For online LC-MS analysis, the nanoUPLC was coupled to the mass spectrometer through a nano-electrospray ionization (nanoESI) source emitter. LC-QTOF-MS/MS Analysis (DDA).

Data dependent analysis were performed on an SYNAPT G2-Si high-definition mass spectrometer (Waters) equipped with a Nano-LockSpray dual electrospray ion source (Waters). Precut fused silica PicoTipR Emitters for nanoelectrospray, outer diameters: 360 μm; inner diameter: 20 μm; 10 μm tip; 2.5" length (Waters) were used for samples and Precut fused silica TicoTipR Emitters for nanoelectrospray, outer diameters: 360 μm; inner diameter: 20 μm; 2.5" length (Waters) were used for the lock mass solution. The eluent was sprayed at a voltage of 2.8 kV with a sampling cone voltage of 25 V and a source offset of 30 V. The source temperature was set to 80 °C. The cone gas flow was 20 liters/h with a nano-flow gas pressure of 0.4 bar and the purge gas was turned off. The SYNAPT G2Si instrument was operated in DDA (data-dependent mode), automatically switching between MS and MS2. Full scan MS and MS2 spectra (m/z 50–2000) were acquired directly after injection to 35 min in resolution mode (20,000 resolution FWHM at m/z 400) with a scan time of 0.1 sec. Tandem mass spectra of up to 10 precursors were generated in the trapping region of the ion mobility cell by using a collision energy ramp from 17/19 V (low mass, start/end) to up to 65/75 V (high mass, start/end). Charged ions (+2, +3, +4, +5, +6) were selected for MS/MS fragmentation across an m/z range of 50 to 2000 with a scan time of 0.25 seconds. For post-acquisition lock mass correction, the doubly charged monoisotopic ion of [Glu1]-fibrinopeptide B at 100 fmol/μl was introduced via the nanoESI source's reference sprayer at a flow rate of 0.5 μl/min and a frequency of 30 seconds into the mass spectrometer. Data processing was performed using MassLynx V4.1 (Waters). Spectra for each sample were extracted from the Total Ion Chromatogram (TIC) and analyzed for masses of interest.

### Pseudotyped SARS-CoV-2 production and Infection

HEK 293 T was grown in DMEM medium supplemented with 10% fetal bovine serum (FBS) with penicillin (100 U mL$^{-1}$) and streptomycin (100 μg mL$^{-1}$) (Gibco, Fisher Scientific, UK) in a humidified incubator with 5% $CO_2$ at 37 °C. 293 T cell line was used to produce pseudotyped VSV-SARS-CoV-2. The subconfluent HEK 293 T cells in a T75 flask were transfected with 15 μg of the pCAG_SARS_2Sdel18 (a generous gift from Stefan Pöhlmann) expression plasmid using Lipofectamine LTX (Invitrogen, MA, USA) according to the manufacturer's protocol. After 8 hours, cells were washed 3 times with DPBS and incubated with VSV-ΔG pseudovirus complemented in trans with the G glycoprotein and bearing a FLuc gene. Cell supernatants containing S protein-pseudotyped VSV were harvested 48 h later, clarified by low-speed centrifugation (900 × g for 10 min), and characterized or stored at −80 °C for later analysis.

A459-ACE2-TMPRSS2 cells were grown in RPMI 1640 medium supplemented with 10% FBS with penicillin (100 U mL$^{-1}$) and streptomycin (100 μg mL$^{-1}$) (Gibco) in a humidified incubator with 5% $CO_2$ at 37 °C. To check the pseudotyped SARS-CoV-2 infection in the presence of Cho inhibitor, the A459-ACE2-TMPRSS2 cell was confluent in 96 well plates. A459-ACE2-TMPRSS2 cells were pre-treated with different concentrations of MβCD for 1.5 h and then incubated with SARS-CoV-2 S protein-bearing pseudoviruses at 37 °C in a humidified atmosphere of 5% $CO_2$ for 4 h. Subsequently, the unbound pseudoviruses were removed by washing with DPBS, and cells were incubated in fresh DMEM medium with 10% FBS, penicillin, and streptomycin for 24 h. For the exogenous-cholesterol supplement assay, A459-ACE2-TMPRSS2 cells were pre-treated with different concentrations (0.5 mM) of MβCD for 1 h, and then supplied with exogenous cholesterol (15 mM) for 1 h. The cells were then incubated with pseudotyped VSV-SARS-CoV-2 at 37 °C in a humidified atmosphere of 5% $CO_2$ for 4 h. The pseudotyped SARS-CoV-2 intensity activity was detected and analyzed using Typhoon (ImageQuantTL software).

### Reporting summary

Further information on research design is available in the Nature Portfolio Reporting Summary linked to this article.

### Data availability

The Source data underlying Figs. 1e–h; 2a–d; 3a–c; 4b–c; 5a–d, f; 6b, d and Supplementary Figs. S1, 2, 4, 5, 7–9 are provided as a Source Data file. The uncropped scan from Fig4a is provided as a Source data file. The AFM raw data generated in this study have been deposited on FigShare [https://doi.org/10.6084/m9.figshare.27613464]. MS data are available via ProteomeXchange with identifier PXD057533. All other relevant data are available from the corresponding authors upon request. Source data are provided with this paper.

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

## Acknowledgements

This work was supported by the Université catholique de Louvain and the Fonds National de la Recherche Scientifique (F.R.S.-FNRS) under the Excellence of Science (EOS) program (Grant ID 40007527 (D.A.)) and the FNRS-Welbio (Grant # CR-2019S-01 (D.A.)). The funders had no role in study design, data collection and analysis, decision to publish, or preparation of the manuscript. Q.Z. and D.A. are grateful to The China Scholarship Council. D.A. is a senior research associate at the F.R.S.-FNRS. The computational part of this work was supported by the National Science Foundation under Grant NSF MCB-2143787 (R.C.B.). The authors thank Hervé Degand for his help with mass spectrometry data acquisition.

## Author contributions

D.A., Q.Z., and G.M.D. conceived/optimized the project. Q.Z conducted all experiments. A.R. performed mass spectrometry experiments with Q.Z. took part in analyzing MS and BLI data. K.D produced the pseudo-typed SARS-CoV-2 virus. A.R. and K.D. performed cholesterol depletion assays for viral infectivity. R.S.L.R. performed the modeling and prepared all MD simulations. R.C.B. and R.S.L.R. performed all the simulations and their analysis. All authors wrote the manuscript.

## Competing interests

The authors declare no competing interests.
