## [Transparent Peer Review file · Nature Communications]

Probing SARS-CoV-2 Membrane Binding Peptide via Single-Molecule AFM-based Force Spectroscopy

Corresponding Author: Professor David Alsteens

Version 0:

Reviewer comments:

Reviewer #1

(Remarks to the Author)

Zhang et al. use single molecule atomic force microscopy (SM-AFM) to investigate the interaction between the SARS-CoV-2 fusion domain and the membrane. Overall, the conclusions of this paper are intriguing and fascinating. The authors characterized both mature (primed by TMPRSS2 enzyme) and premature versions of the fusion peptide, as well as the role of its conserved disulfide bridge in membrane interaction. Using two distinct membrane compositions, they showed that the fusion peptide preferentially binds to cholesterol-enriched membranes over those containing sphingomyelin, and cholesterol depletion from the host membrane significantly reduces SARS-CoV-2 infectivity.

The main result was the use of SM-AFM to monitor SARS-CoV-2 FP binding to the lipid membrane. Some secondary findings include a fivefold quicker binding of FP1 to cholesterol-containing vesicles than to sphingomyelin-containing vesicles, microsecond-long MD simulations, and so on. Although the content is typically technically sound and the meaning is clear, the authors might consider making some changes to the presentation of scientific facts to improve readability and comprehension. Nevertheless, I am unable to provide commentary on the data interpretation and correlation utilizing the Dudko-Hummer-Szabo (DHS) model due to my limited knowledge of the scientific literature and its practical implementation. This evaluation should be conducted by another reviewer or reviewers (Figure 5 d and f).

Although the work is remarkable, careful, and of broad appeal, certain specific elements require more explication and/or revision:

The main topic is whether cholesterol (Chol) adsorption influences the kinetics of SARS-CoV-2 FP binding to the lipid membrane, resulting in the probable artifact observation. The authors should discuss this point. Perhaps they could perform some control experiments or have some ideas.

Another key question is the validity of the line-scan method for the TMPRSS2-FP. I was left wondering whether AFM line scans offered as trustworthy and interpretable data as the full AFM movies. From what I understand, line-scan relies on the system's stability to record the same line (spatial position) throughout time. The authors should provide a complete kymograph analysis in their revision.

Another issue is the cholesterol lipid composition tests. Perhaps the author should conduct further control studies in the AFM field using a different common lipid composition (DOPC/DPPC/Chol=40:40:20).

Another problem is the microsecond-long MD simulations. It would be really beneficial if the authors could add another popular program, Amber, ACEMD3, or Amberff14SB force field in their revision.

Reviewer #2

(Remarks to the Author)

The authors investigated the interaction between the SARS-CoV-2 Spike protein's fusion peptide and the host membrane using both in vitro and in silico experiments. They found that the fusion peptide exhibits a higher affinity for cholesterol-enriched membranes. This finding provides insights into the membrane fusion process. However, I have several concerns and comments regarding the manuscript:

1. The authors referred to the fusion peptide region as consisting of 40 amino acids (residues 816-855). This may not be accurate, as the exact position of the fusion peptide region was not well established until the publication of a cryo-EM structure of the SARS-CoV-2 postfusion S trimer last year. This structure revealed that residues 834-856 constitute the fusion peptide proximal region (FPPR), which is partially embedded in the membrane, while residues 816-834 are located outside the membrane at a later stage. I suggest using a different name for this region to avoid confusion.
2. The designed FP1 and FP2 peptides may not accurately mimic the behavior of the fusion peptide in the spike protein, especially the FPPR region, which is known to adopt different conformations among spike variants. The difference in membrane-binding response between FP1 and FP2 to various lipids may originate from the peptides themselves. The manuscript lacks experiments demonstrating that FP1 and FP2 fold correctly. Additional experiments are needed to support this conclusion, including positive and negative control experiments to rule out artifacts.
3. Regarding the S2 subunit experiment, the authors did not specify which type of S2 protein was used. If the S2 subunit protein was not modified, it would not remain in the prefusion state and would quickly transition to the postfusion state. Therefore, enzymatic experiments would induce different states in FP1 and FP2, potentially undermining the conclusions. Have the authors observed differences in S2 subunit states before and after TMPRSS2 treatment?
4. The authors used Methyl- β -cyclodextrin (M β CD) to investigate the impact of cholesterol depletion on viral infection. Control experiments are needed to support this conclusion. Additionally, cholesterol depletion affects membrane features such as fluidity, which could influence the conclusions drawn.
5. The authors' system design is intriguing, and the observation of interactions between FP1 and the membrane, albeit without insertion into the membrane, is notable. Further work could explore the roles of other regions in the membrane fusion process to better understand which region is responsible for membrane insertion.

Reviewer #3

(Remarks to the Author)

The authors investigate the interaction of the SARS-CoV-2 spike protein's fusion peptide (FP) with host membranes, revealing a preferential affinity for cholesterol-enriched membranes and the critical role of the FP's disulfide bridge in stabilizing membrane interaction. However, over the past one to three years, researchers have devoted considerable effort to exploring the fusion mechanism between SARS-CoV-2 and host cells, as evidenced by studies published in various scientific journals, such as *Journal of the American Chemical Society* 2021, 143, 33, 13205-13211; *Biochemistry* 2023, 62, 21, 3033-3035; *Journal of Chemical Information and Modeling* 2021, 61, 1, 423-431; *Biophysical Journal* 2021, 120, 2914-2926; *Journal of Molecular Biology* 2022, 434, 167280, and more. While the field has made strides, the manuscript in question appears to follow this trend without introducing significant innovation.

Major comments:

The themes addressed, the research methods employed (including molecular dynamics simulations), and the conclusions drawn in the manuscript all appear to lack major innovation. The authors should, therefore, highlight the uniqueness and innovative aspects of their research in order to capture the attention of readers.

As mentioned in the last three lines of the introduction: "These insights emphasize the crucial role of FP-membrane binding as a promising therapeutic target for developing broadly effective therapies against coronaviruses", this is not a novel conclusion.

Additionally, the current study is limited to SARS-CoV-2 and fails to consider other viral mutants, which restricts the breadth and depth of the research to a certain extent (*Nature Communications* 2024, 15, 4056). To gain a more comprehensive and in-depth understanding of the fusion mechanisms of coronaviruses, the authors may consider incorporating more mutants into their future research scope.

Minor comments:

1. As mentioned in lines 110-111: "we designed two peptides with slight modifications, FP1 and FP2". Please clarify the design rationale for these two peptides.
2. The basis for constructing the membrane model needs to be further elucidated, including but not limited to the proportion of cholesterol, as the composition of the membrane seems to be related to an important conclusion in the manuscript.
3. As mentioned in line 218: "FP1 and FP2 structures, obtained using AlphaFold2", different software tools should be used to verify the prediction results of AlphaFold2 to ensure the accuracy of the initial structure used for simulation.
4. According to lines 221-226, for molecular dynamics simulations, observing protein insertion into the cell membrane depends on a variety of factors beyond simulation time, such as the accuracy of the model and the initial conditions set, like temperature, etc. (*PLoS one* 2012, 7, e47596). Therefore, the contents of this part are recommended to modify accordingly.
5. Based on the results in Fig. 3d, please explain the reason for the decrease in average contact in longer molecular dynamics simulations. Is this limited by the fact that the authors only used the FP structure for their simulations instead of the full-length Spike protein?
6. Fig. 2 and Fig. 4 are ill-labeled and unsightly, which requires further revision.

Reviewer #4

(Remarks to the Author)

The paper titled "Probing SARS-CoV-2 Fusion Domain-Membrane Interaction via Single-Molecule AFM-based Force Spectroscopy" investigates the binding between the SARS-CoV-2 spike protein's fusion peptide (FP) and host cell membranes. Employing both in vitro and computational approaches, the study reveals a distinct preference of the FP for cholesterol-rich membranes, emphasizing the crucial role of the internal disulfide bridge in stabilizing this interaction. This research offers significant insights into the mechanisms of viral membrane fusion, presenting potential targets for therapeutic intervention. Notably, the integration of single-molecule AFM-based force spectroscopy with computational simulations enhances the robustness of the findings.

Specific comments on data analysis of Figure 5:

The use of single-molecule force spectroscopy in constant-force mode to derive force-dependent lifetimes is meticulously analyzed using Bell's formula and its generalization (DHS model) under Kramers theory. This dual analytical method validates the findings, as shown by the comparable kinetic parameters between the two models. The execution of this comparison in Figure 5 is commendable, reinforcing the reliability of the results. However, further discussion on the scaling factor in the DHS model would be beneficial, particularly regarding how it affects the accuracy of the derived free-energy parameters. Addressing this would clarify the potential variability and enhance the robustness of the findings, thereby improving the overall impact of the study.

Minor corrections:

Fig 4c. Correct typo 'Bell modle Fit'

Legend of Fig 5b.

Pls specify the number of binding events analyzed to determine the frequencies.

Legend of Fig 5d.

- Explain colored data points, are they averages or mean? What are the error bars. Apparently the data has been binned, which bin has been applied?
- The authors write 'Those data points were fitted using the DHS model.' Which data points the binned colored ones or the grey ones?

Legend of Fig 5f.

- The title of the legend needs correction. I think 'binds' needs to be replaced by 'binding'.
- The authors write 'The kinetics and thermodynamics analysis reveal a binding' pls specify analysis of what?
- The authors write 'can be explain' pls write 'can be explained'

Version 1:

Reviewer comments:

Reviewer #1

(Remarks to the Author)

This manuscript is a resubmission and the authors have successfully addressed major concerns raised previously.

Reviewer #2

(Remarks to the Author)

I would like to thank the authors for their detailed responses and the additional experiments they have conducted. The revisions have addressed most of my concerns, and I am largely satisfied with the improvements made to the manuscript. However, there remains one major issue related to the S2 subunit that requires further clarification. Below are my specific comments:

1. The renaming of the fusion peptide region to "membrane-binding peptide" (MBP) is a sensible change given the recent cryo-EM data, and I appreciate the authors' effort to avoid confusion.
2. The additional MD simulations and control experiments, particularly these involving the Arg846 mutation and MPP peptide, greatly strengthen the manuscript. However, I would suggest that structure prediction experiments also be performed on the Arg846 mutation to ensure that this mutation does not alter the peptide's structure. AlphaFold is a reasonable approach for such predictions, but the MBP region appears highly variable across different SARS-CoV-2 Spike protein variants. Moreover, the AlphaFold-predicted structure deviates slightly from published structures. To further solidify these finding, it would be valuable to confirm the structural integrity using experimental techniques such as HPLC if feasible within a reasonable timeframe.
3. While the authors have provided additional information regarding the S2 subunit used in their experiments, one key concern remains. The S2 subunit was purchased from a commercial source and is maintained in a soluble state. Without any modification to stabilize this subunit, it is highly likely that the S2 subunit remains in the post-fusion conformation. This point has not been sufficiently addressed in the manuscript, and the figure depicting the S2 subunit adds to the confusion. It raises concerns about the conclusions drawn from these experiments. If necessary, additional experiments or more detailed explanation should be provided.
4. The new control experiments on cholesterol depletion and repletion provide strong support for the conclusions. I appreciate the clarity of the results showing that viral infection increases following cholesterol repletion. It would be more

helpful to clarify the potential role of the actual fusion peptide during these experiments.

5. The authors' observations regarding the interactions between MBP1 and the membrane are intriguing and open new avenues for future research. I appreciate the authors' consideration of this point.

In conclusion, I am largely satisfied with the revisions, but the issue concerning the S2 subunit's structural state remains unresolved. I would recommend addressing this point in detail to ensure the robustness of the conclusions drawn from the S2 subunit experiments. Once this is resolved, I believe the manuscript will be suitable for publication.

Reviewer #3

(Remarks to the Author)

In this round, the authors have implemented a series of revisions and responses to the previous review comments. However, despite the authors' efforts in addressing the review comments, the fifth point in Minor comments has not received sufficient attention. I have concerns about the role of the Spike protein S2 subunit in the virus-host cell membrane fusion process. Although the manuscript primarily focuses on the interaction between the membrane-binding peptide (MBP) and the lipid membrane, the effect of S2 on cell membrane fusion following TMPRSS2 cleavage should not be overlooked, whether in vitro or in silico experiments. In addition, some of the figures are still not satisfactory and need further revision.

Reviewer #4

(Remarks to the Author)

The authors have addressed all of my comments satisfactorily. The revised manuscript has been much improved and is a pleasure to read.

Version 2:

Reviewer comments:

Reviewer #2

(Remarks to the Author)

The authors have addressed all of my comments. The revised manuscript has been much improved and is a pleasure to read.

Reviewer #3

(Remarks to the Author)

Thank the authors for providing additional explanations to my concerns, and I have no further comments.

made.

Point-by-Point Response to the Reviewers' Comments

Reviewer #1

Zhang et al. use single molecule atomic force microscopy (SM-AFM) to investigate the interaction between the SARS-CoV-2 fusion domain and the membrane. Overall, the conclusions of this paper are intriguing and fascinating. The authors characterized both mature (primed by TMPRSS2 enzyme) and premature versions of the fusion peptide, as well as the role of its conserved disulfide bridge in membrane interaction. Using two distinct membrane compositions, they showed that the fusion peptide preferentially binds to cholesterol-enriched membranes over those containing sphingomyelin, and cholesterol depletion from the host membrane significantly reduces SARS-CoV-2 infectivity.

The main result was the use of SM-AFM to monitor SARS-CoV-2 FP binding to the lipid membrane. Some secondary findings include a fivefold quicker binding of FP1 to cholesterol-containing vesicles than to sphingomyelin-containing vesicles, microsecond-long MD simulations, and so on. Although the content is typically technically sound and the meaning is clear, the authors might consider making some changes to the presentation of scientific facts to improve readability and comprehension.

Authors: We thank the reviewer for his/her comments on our manuscript and are glad to know that he/she found our manuscript intriguing and fascinating. We have now addressed the questions below and made some changes to the manuscript to improve readability and comprehension.

Nevertheless, I am unable to provide commentary on the data interpretation and correlation utilizing the Dudko-Hummer-Szabo (DHS) model due to my limited knowledge of the scientific literature and its practical implementation. This evaluation should be conducted by another reviewer or reviewers (Figure 5 d and f).

Authors: We agree that this is an important point of the manuscript. We would like to emphasize that the Dudko-Hummer-Szabo (DHS) model, which is derived from the Bell model (Dudko et al., Proc. Natl. Acad. Sci., 2008) (1), is commonly used to interpret single-molecule data from AFM (see a recent review on the different models: Müller et al., Chem. Rev., 2021) (2) and optical tweezers (Bustamante et al., Nat. Rev. Methods Primer, 2021) (3). Therefore, we believe that this model was perfectly suited to interpret our experimental data, as also pointed out by another reviewer.

Although the work is remarkable, careful, and of broad appeal, certain specific elements require more explication and/or revision:

The main topic is whether cholesterol (Chol) adsorption influences the kinetics of SARS-CoV-2 FP binding to the lipid membrane, resulting in the probable artifact observation. The authors should discuss this point. Perhaps they could perform some control experiments or have some ideas.

Another key question is the validity of the line-scan method for the TMPRSS2-FP. I was left wondering whether AFM line scans offered as trustworthy and interpretable data as the full AFM movies. From what I understand, line-scan relies on the system's stability to record the same line (spatial position) throughout time. The authors should provide a complete kymograph analysis in their revision.

Authors: We would like to combine these two comments together because they are both related to the feasibility and the efficacy of experiment set-up.

Fig. R1. Set-up of AFM-based SMFS in height clamp mode with either the AFM tip functionalized with a peptide (left) or with the S2 protein (right).

First, we would like to clarify our experimental setup. As shown in **Fig. R1** (**Fig. 2a,b**, and **Figs. S2, S3**), the AFM tips were functionalized with either a peptide or the S2 protein, while the supported lipid bilayer was adsorbed onto a bare mica surface. Before the AFM tip engages, the peptide on the tip and the lipid bilayer are not in contact. During height-clamp force experiments, which are state-of-the-art AFM techniques, the AFM tip is positioned several nanometers away from the lipid bilayer (4–6 nm in our case). At this distance, and based on the length of the PEG linker and the peptide/protein, the peptide or protein is free to interact with the lipid membrane. Upon interaction, the AFM tip bends due to the force acting between the peptide/protein and the lipid bilayer. The bending of the cantilever, and thus the force, is recorded over time and displayed as Force vs. Time curves (**Fig. 2c,d** and **Supporting Movie**). We have now added a few lines to the main text to clarify the setup (lines 183–187). “The functionalized AFM tip was positioned 4–6 nm above the lipid bilayer, allowing specific interactions between the MBP and the lipid layer to be captured as Force vs. Time (FT) curves (see **Supporting Movie**). The resulting force (in pN) between the functionalized tip and the lipid bilayer was continuously monitored over a 10-second period, during which the MBP and the lipid bilayer interacted.”

Additionally, we performed control experiments to further validate our findings. Based on our MD simulations, which indicated that Arg846 in the MBP1 peptide sequence is crucial for establishing initial contact with the lipid bilayer, we synthesized a single-point mutation peptide as a negative control. Our results showed that the MBP1 mutant did not bind to the lipid bilayer (Fig. R2, see Fig. S4). Furthermore, we observed that the S2 protein treated with TMPRSS2 binds to the DOPC/Chol membrane, whereas no binding events were detected for the untreated S2 protein. Collectively, these results confirm the specificity of our binding events, demonstrating that they specifically arise from the functional state of the peptide or protein.

Fig. R2. Plot showing quantification of specific binding events of different peptides with the DOPC/Chol membrane.

Another issue is the cholesterol lipid composition tests. Perhaps the author should conduct further control studies in the AFM field using a different common lipid composition (DOPC/DPPC/Chol=40:40:20).

Authors: To address this comment, we conducted an additional BLI experiment. MBP1 was covalently attached to an amine-reactive biosensor, similar to the setup in Fig. 1. Lipid vesicles containing 40% DOPC, 40% DPPC, and 20% Chol at a concentration of 1 mM were added to the biosensor to measure the association and dissociation rates with MBP1. The results showed that the binding rate of MBP1 to DOPC/DPPC/Chol (40:40:20) vesicles is 0.018 nm/s (Fig. R3, see Fig. S1), which is consistent with the binding rate of MBP1 to DOPC/Chol (70:30, 0.017 nm/s, Fig. 1e). These findings suggest that the DOPC/Chol (70:30) lipid composition used in our AFM measurements is relevant and that the differences in lipid ratios between the two experimental setups have minimal influence on the overall binding rate. We have included this result in Supplementary Information Fig. S1, and it is described in the manuscript as follows

(lines 134-139): “To extend the interaction to more complex lipid compositions, we tested the binding rate of MBP1 to a mixed lipid composition (DOPC/DPPC/Chol = 40:40:20). We found that the binding rate of MBP1 to DOPC/DPPC/Chol (40:40:20) vesicles is 0.018 nm/s (Fig. S1), similar to the rate observed for DOPC/Chol (70:30) vesicles (0.017 nm/s). This consistency underscores the predominant role of Chol in enhancing binding kinetics, rather than the contributions from other lipid components.”

Fig. R3. BLI sensorgrams showing the real-time binding interaction between SARS-CoV-2 MBP1 and lipid vesicles composed of DOPC/DPPC/Chol in a 40:40:20 ratio.

Another problem is the microsecond-long MD simulations. It would be really beneficial if the authors could add another popular program, Amber, ACEMD3, or Amberff14SB force field in their revision.

Authors: We appreciate the reviewer's comment regarding the use of different MD simulation programs and force fields. However, we respectfully disagree with the suggestion to incorporate Amber, ACEMD3, or the Amberff14SB force field into our study.

The CHARMM Force Field is a well-established tool in the field of molecular dynamics, widely recognized and extensively validated by the scientific community. Its robustness is evidenced by its widespread adoption and the numerous citations it has garnered over the years. We believe that switching to the Amber Force Field would not provide any additional scientific benefit, as modern force fields, including CHARMM and Amber, typically yield comparable statistical results across a wide range of systems.

Regarding the suggestion to use Amber or ACEMD3 instead of NAMD, we would like to highlight the significant strengths of NAMD and its accompanying visualization software, VMD. NAMD and VMD have over 150,000 registered users. On average, this software receives a new citation every 55 minutes, underscoring their reliability and popularity. NAMD alone has over 25,000 citations in scientific articles, with over 1,000 just in 2024. Both of these softwares are developed by the NIH Center for Macromolecular Modeling and Visualization and supported by an NIH grant, for which one of the authors (RCB) is a co-PI. NAMD is frequently employed in

public supercomputing centers and serves as the “acceptance test” for most new supercomputers in the US, a distinction not achieved by any other MD software.

Additionally, our study involves over 25 microseconds of MD simulation of a system with just over 300,000 atoms. The MD protocol used in our work would require approximately 50 days of computation on a DGX-A100 node, costing nearly \$20,000. Undertaking such extensive calculations without a clear scientific rationale would be unjustifiable.

In conclusion, while we acknowledge the reviewer's perspective, we believe that the CHARMM Force Field and NAMD are well-suited for our study and that switching to another MD software would not yield any significant scientific advantage. We hope this clarifies our position and the rationale behind our choices.

Reviewer #2 (Remarks to the Author)

The authors investigated the interaction between the SARS-CoV-2 Spike protein's fusion peptide and the host membrane using both in vitro and in silico experiments. They found that the fusion peptide exhibits a higher affinity for cholesterol-enriched membranes. This finding provides insights into the membrane fusion process.

Authors: We thank the reviewer for the favorable feedback on our work. The insightful comments have significantly helped us to improve the manuscript and supporting information.

However, I have several concerns and comments regarding the manuscript:

1. The authors referred to the fusion peptide region as consisting of 40 amino acids (residues 816–855). This may not be accurate, as the exact position of the fusion peptide region was not well established until the publication of a cryo-EM structure of the SARS-CoV-2 postfusion S trimer last year. This structure revealed that residues 834–856 constitute the fusion peptide proximal region (FPPR), which is partially embedded in the membrane, while residues 816–834 are located outside the membrane at a later stage. I suggest using a different name for this region to avoid confusion.

Authors: We agree with the reviewer that the recent cryo-EM analysis demonstrates that residues 834–856 are partially embedded in the membrane, while residues 816–834 are positioned outside the membrane at the post-fusion stage (Shi et al., Nature, 2023) (4). Since our peptide sequence spans residues 816 to 855 and is associated with lipid binding, we have decided to rename the peptide as the membrane-binding peptide (MBP).

2. The designed FP1 and FP2 peptides may not accurately mimic the behavior of the fusion peptide in the spike protein, especially the FPPR region, which is known to adopt different conformations among spike variants. The difference in membrane-binding response between FP1 and FP2 to various lipids may originate from the peptides themselves. The manuscript lacks experiments demonstrating that FP1 and FP2 fold correctly. Additional experiments are needed to support this conclusion, including positive and negative control experiments to rule out artifacts.

Authors: We appreciate the reviewers' insightful question regarding the conformation and correct folding of our peptides. To address these concerns, we conducted MD simulations using AlphaFold to predict the peptide structures, which revealed that the peptides fold into stable conformations with high IDDT scores (see Fig. S5). During simulations in the microsecond range, the peptides maintained stable structures. We have incorporated this information into the main text on lines 243–247: “The structures predicted by AlphaFold2 had pLDDT scores of 81.4 for MBP1 and 78.4 for MBP2, both of which fall within the 'confident' range according to AlphaFold (4). Notably, the alpha-helix regions exhibited even higher scores, exceeding 90, which is considered 'very high confidence' (Fig. S5). MBP1 and MBP2 structures were equilibrated using the MD protocol and inserted into the same two membrane environments as used previously.”

Additionally, as suggested by the reviewer, we included new data to further validate the specificity of the peptide binding to the lipid bilayer. Our MD simulations indicated that Arg846

in the MBP1 peptide sequence is critical for establishing initial contact with the lipid bilayer. To test this, we synthesized a single-point mutant peptide as a negative control, where Arg846 was mutated to Gly (sequence: SFIEDLLFNKVTLADAGFIKQYGDCLGDIAAGDLICAQKFLPETGG). This mutant was tested using our AFM approach. We observed that the binding probability was significantly reduced by this mutation, demonstrating a strong correlation with the MD simulation and validating our experimental approach.

As a positive control, we synthesized a membrane-penetrating peptide (MPP) from the S protein, identified by cryo-EM as the region that inserts into the host lipid membrane (4). This MPP peptide (residues 866–910) was synthesized with a sortase tag at the C-terminus and a polyG sequence for added flexibility (sequence: ⁸⁶⁶TDEMIAQYTSALLAGTITSGWTFGAGAALQIPFAMQMAYRFNGI⁹¹⁰GGGGGLPETGG).

Our results showed that this MPP peptide binds to the lipid bilayer with a high binding frequency (Fig. R4), even higher than that of MBP1, which aligns well with the structural data obtained by cryo-EM. These findings have been added to Supplementary Information Fig. S4 and are described in the manuscript as follows: “As a positive control, a membrane-penetrating peptide (MPP) of the S protein (residues 866–910), validated by cryo-EM as the region that crosses the host membrane (4), binds to the DOPC/Chol membrane with a 16.5% binding frequency (Fig. S4).” (lines 197-199). Additionally, on lines 381-384: “To further confirm that the Arg846 residue is directly involved in membrane interaction, we used a point mutant peptide (MBP1 mutant, Arg846Gly) in the same AFM-based SMFS experiment. Strikingly, the results showed that the MBP1 mutant did not bind to the DOPC/Chol membrane (Fig. S4), highlighting the critical role of Arg846 in direct membrane association.”

Fig. R4. Plot showing the quantification of specific binding events between different peptides tethered on the AFM tip and the DOPC/Chol bilayer.

3. Regarding the S2 subunit experiment, the authors did not specify which type of S2 protein was used. If the S2 subunit protein was not modified, it would not remain in the prefusion state and would quickly transition to the postfusion state.

Therefore, enzymatic experiments would induce different states in FP1 and FP2, potentially undermining the conclusions. Have the authors observed differences in S2 subunit states before and after TMPRSS2 treatment?

Authors: We apologize for the lack of detail regarding the S2 subunit. The S2 subunit (residues Ser686–Lys1211) used in our experiments was obtained from R&D Systems. This would allow us to generate the S2' by TMPRSS2 enzymatic cleavage, hence exposing the membrane-binding region. The protein sequence corresponds to the original SARS-CoV-2 Spike protein from the early isolate of the original strain (USA/WA1/2020, Wuhan-1, Accession#: YP_009724390, collected on December 24, 2019). This information has been clarified in the Methods section: “Wuhan-1, Accession#: YP_009724390, collected on December 24, 2019, R&D Systems.”

In our original submission, we found that the S2 subunit treated with TMPRSS2 binds to the DOPC/Chol membrane (9 binding events observed over 339 force-time curves recorded), whereas no binding events were detected for the untreated S2 subunit (no binding events observed over 453 force-time curves recorded). These results suggest that TMPRSS2 treatment induces the S2 subunit into a state with an exposed membrane-binding region. In the revised manuscript, we have rephrased our findings to further clarify the S2 subunit's binding to the lipid membrane after TMPRSS2 treatment and have strengthened our overall conclusions. Furthermore, successful TMPRSS2 cleavage of S2, which was verified by western blot, confirms that the S2 subunit was originally not modified and harbored a prefusion state in a original fold since served as a substrate for the TMPRSS2 enzyme.

4. The authors used Methyl- β -cyclodextrin (M β CD) to investigate the impact of cholesterol depletion on viral infection. Control experiments are needed to support this conclusion. Additionally, cholesterol depletion affects membrane features such as fluidity, which could influence the conclusions drawn.

Authors: We thank the reviewer for the valuable suggestion. To address the concerns raised, we performed an additional control experiment. Prior to VSV-SARS-CoV-2 infection, A549-ACE2-TMPRSS2 cells were treated with Methyl- β -cyclodextrin (M β CD) to deplete cholesterol, and then exogenous cholesterol was added. The results showed that viral infection increased more than twofold after the addition of exogenous cholesterol to M β CD-treated A549-ACE2-TMPRSS2 cells. In contrast, the addition of exogenous cholesterol led to only a slight increase in viral infection in cells that were not treated with M β CD (Fig. R5). These findings indicate that exogenous cholesterol supplementation restores viral infection in cholesterol-depleted cells, supporting our conclusion that cholesterol content in the host membrane significantly influences SARS-CoV-2 infection.

In addition, we would like to point out that the plasma membrane of living cells is highly heterogeneous, and the local concentration of cholesterol can vary, affecting membrane properties such as fluidity. This intrinsic variability in cell membranes can influence experimental outcomes. However, the fact that our infectivity assays demonstrate a reversible

increase in infection following cholesterol depletion and subsequent repletion suggests that the cell physiology has not been irreversibly affected. This observation further supports the relevance of cholesterol in modulating SARS-CoV-2 infection. Additionally, our AFM data show significant differences in binding forces and bond lifetimes between lipid bilayers containing cholesterol and those containing only sphingomyelin.

Fig. R5. The exogenous-cholesterol supplement assay, A549-ACE2-TMPRSS2 was treated with MβCD (0.5 mM) for 1h, and then supplied with exogenous cholesterol (15 mM) for 1h. The virus infection is normalized in percent by negative control.

These new data have been added to the Figure 6 and the results have been described on lines 423-433: “To further confirm the cholesterol-dependence of VSV-SARS-CoV-2 infection, A549-ACE2-TMPRSS2 cells were treated with Methyl-β-cyclodextrin (MβCD) to deplete cholesterol and subsequently supplemented with exogenous cholesterol. The results showed that SARS-CoV-2 infection increased more than twofold following cholesterol repletion in MβCD-treated cells. This indicates that cholesterol depletion significantly impairs viral entry, which can be effectively restored by adding exogenous cholesterol. In contrast, in cells that were not treated with MβCD, the addition of exogenous cholesterol resulted in only a slight increase in infection (Fig. 6c and d). These findings suggest that cholesterol content in the host cell membrane plays a critical role in enhancing SARS-CoV-2 infection, likely by increasing the binding affinity of viral membrane-binding peptide to host membrane.”

5. The authors' system design is intriguing, and the observation of interactions between FP1 and the membrane, albeit without insertion into the membrane, is notable. Further work could explore the roles of other regions in the membrane fusion process to better understand which region is responsible for membrane insertion

Authors: We appreciate the reviewer’s positive feedback on our system design and the observations regarding the interactions between MBP1 and the membrane. The lack of insertion into the membrane is indeed intriguing and warrants further investigation. Future work could explore the roles of other regions of the protein in the membrane fusion process

to better understand which regions are responsible for membrane insertion. This could provide deeper insights into the mechanisms underlying membrane fusion and identify specific domains that facilitate this critical process. Additionally, it would be valuable to investigate whether this process is mediated by other membrane proteins, as has been proposed for other viruses known to enter human cells intact.

SARS-CoV-2 must overcome kinetic barriers in both the virus and host cell membranes for effective membrane binding. The necessary free energy is derived from the refolding of the S protein during the transition from the pre-fusion to the post-fusion state, which involves interactions between membrane-interacting elements of the S protein (including residues 816–855 and 866–909) and the host membrane. Our work focuses on the membrane-binding segment (residues 816–855), a conserved domain targeted by several broadly neutralizing antibodies (5, 6).

This data is particularly relevant for the development of vaccines and therapeutics that address emerging SARS-CoV-2 variants and potential future coronaviruses. The peptide spanning residues 866–909 of the S protein, validated by cryo-EM as a membrane-crossing segment (7), serves as a positive control in our studies. Although this peptide is not an efficient vaccine target due to its limited accessibility by antibodies, a comprehensive understanding of all membrane-interacting elements in viral membrane fusion will be crucial for identifying vaccine and therapeutic candidates that target this process and prevent viral infection.

Reviewer #3 (Remarks to the Author)

The authors investigate the interaction of the SARS-CoV-2 spike protein's fusion peptide (FP) with host membranes, revealing a preferential affinity for cholesterol-enriched membranes and the critical role of the FP's disulfide bridge in stabilizing membrane interaction. However, over the past one to three years, researchers have devoted considerable effort to exploring the fusion mechanism between SARS-CoV-2 and host cells, as evidenced by studies published in various scientific journals, such as *Journal of the American Chemical Society* 2021, 143, 33, 13205-13211; *Biochemistry* 2023, 62, 21, 3033–3035; *Journal of Chemical Information and Modeling* 2021, 61, 1, 423–431; *Biophysical Journal* 2021, 120, 2914–2926; *Journal of Molecular Biology* 2022, 434, 167280, and more. While the field has made strides, the manuscript in question appears to follow this trend without introducing significant innovation.

Authors: We thank the reviewer for the feedback and suggestions, which we have incorporated into a revised manuscript.

Major comments:

The themes addressed, the research methods employed (including molecular dynamics simulations), and the conclusions drawn in the manuscript all appear to lack major innovation. The authors should, therefore, highlight the uniqueness and innovative aspects of their research in order to capture the attention of readers.

Authors: Thank you for this important suggestion, which provides an opportunity to emphasize the innovation in our research. We have revised the discussion section to include a brief literature review that highlights the novel aspects of our manuscript.

Previous studies have suggested that the membrane-binding peptide (MBP) attaches quickly to cellular membranes without completely crossing the lipid bilayer (8-10), which aligns with our findings. However, these reports typically rely on limited experimental methods or *in silico* approaches. Our study distinguishes itself by employing a range of complementary techniques—AFM, BLI, MD simulation, and viral infectivity assays—to confirm MBP binding to the lipid bilayer both *in vitro* and *in silico*.

Notably, we utilized state-of-the-art AFM-based single-molecule force spectroscopy (SMFS) in height-clamp mode to quantitatively analyze the thermodynamics of MBP binding to the membrane. By employing site-directed enzyme-assisted techniques, we also demonstrated the role of the TMRSS2-primed N-terminal region of the S2 subunit of the spike protein in membrane attachment. Additionally, mass spectrometric analyses (ESI positive mode) provided direct qualitative evidence of the oxidative state of the Cys residues in solution, which informs the chemical nature of cysteines on the AFM tip.

For the first time, we experimentally quantified the binding kinetics and thermodynamics of MBP to the membrane and provided evidence of the role of the internal conserved disulfide bridge in the MBP domain in establishing the strong binding necessary for SARS-CoV-2

membrane fusion. This work offers direct insights into the role of cysteine residues in SARS-CoV-2 infectivity and demonstrates how local oxidation at a single pair of cysteines can globally impact protein conformation and binding. Importantly, a single point mutation (Arg846) in MBP is sufficient to prevent its attachment to the membrane, presenting a potential target for inhibiting SARS-CoV-2 infection. We also found that MBP enriched with cholesterol binds to membranes more effectively, while cholesterol depletion significantly impairs SARS-CoV-2 infection.

We have incorporated this information into the discussion section of the revised manuscript on lines 465-:

“Previous reports have suggested that MBP attaches rapidly to cellular membranes without fully crossing the lipid bilayer (8-10), consistent with our findings. However, to our knowledge, for the first time at the single peptide level, we experimentally quantified the binding kinetics and thermodynamics of MBP to the membrane and demonstrated the role of the internal conserved disulfide bridge in the MBP domain in establishing the strong binding necessary for SARS-CoV-2 membrane fusion. Furthermore, the single point mutation (Arg846) in MBP is sufficient to prevent its attachment to the membrane.”

As mentioned in the last three lines of the introduction: “These insights emphasize the crucial role of FP-membrane binding as a promising therapeutic target for developing broadly effective therapies against coronaviruses”, this is not a novel conclusion.

Authors: We agree to the reviewer’s suggestion and re-phrased the revised manuscript on lines 104-107. “These observations were further substantiated under physiologically relevant conditions, demonstrating that cholesterol facilitated binding between the TMPRSS2-cleaved S2 subunit and the membrane, leading to the inhibition of SARS-CoV-2 infectivity in relevant assays.”

Additionally, the current study is limited to SARS-CoV-2 and fails to consider other viral mutants, which restricts the breadth and depth of the research to a certain extent (Nature Communications 2024, 15, 4056). To gain a more comprehensive and in-depth understanding of the fusion mechanisms of coronaviruses, the authors may consider incorporating more mutants into their future research scope.

Authors: We sincerely thank the reviewer for providing this insightful comment, which will be instrumental in guiding our future experiments. We aim to address these aspects in MBP binding to membrane in this report. MBP domain is the high sequence conservation and the targets of multiple broadly antibodies against coronaviruses (5, 6). Our current findings on MBP binding to membrane, particularly essential role of the conserved disulfide bond (stability/folding of the MBP) as well as the Arg846 residue (positive charge of the MBP in this region), provide a solid foundation for future research directions aimed at unraveling all membrane-interacting elements and other viral mutant’s interaction and its role in viral infection.

Minor comments:

1. As mentioned in lines 110-111: “we designed two peptides with slight modifications, FP1 and FP2”. Please clarify the design rationale for these two peptides.

Authors: Thank you for this suggestion. Our objective is to probe the function of the peptide region (MBP1: residues 816-855) exposed by TMPRSS2 in SARS-CoV-2 membrane fusion. So, this sequence is our base sequence. As a negative control, the 11 amino acids is added in MBP2. To investigate an oriented single molecule binding between peptides and membrane, we added a sortase A tag at the C-terminus (residues: LPETGG). Sortase-mediated coupling allows for the attachment of molecules to AFM tips or substrates at specific sites, ensuring that the forces measured are due to interactions at defined points on the molecule. Furthermore, the site-specific attachment of molecules ensures that they are oriented in a defined manner on the surface which enhances the robustness of this experiment. We have now revised and rephrased our manuscript to include the following lines to substantiate our choice of peptide sequences. Lines 113-114 “This peptide is highly conserved across coronaviruses, and targeted by several broadly neutralizing antibodies.”

Line 165-169 “This approach allows for precise control over the grafting geometry, providing a reliable method to specifically probe the interaction at this site (11, 12). This ensures that the experimental setup accurately captures the movement of the exposed N terminus of the MBP and faithfully represents the encounter between SARS-CoV-2 and host cells.”

2. The basis for constructing the membrane model needs to be further elucidated, including but not limited to the proportion of cholesterol, as the composition of the membrane seems to be related to an important conclusion in the manuscript.

Authors: For the used lipids composition in our report, we main aim is to study the function of cholesterol for SARS-CoV-2 infection. Because the PC is the major component of mammalian cell membrane, we used it as main composition of lipid membrane to mimic cell membrane. Sphingomyelin is a type of sphingolipid found in animal cell membranes; we use it as a negative control in our experiment. Indeed, to address the similar concern from other reviewer, we have carried out an additional experiment with more common lipid composition, containing 40% DOPC, 40% DPPC, and 20% Chol. The result shown that the binding rate of MBP1 to DOPC/DPPC/Chol (40:40:20) vesicles is 0.018 nm s^{-1} (Fig. R3, Fig S1) and is consistent with binding rate of MBP1 to DOPC/Chol (70:30, 0.017 nm s^{-1} , Fig. 1e). We believe that the used lipid composition to DOPC/Chol (70:30) for our AFM measurement is very effective. We added the following lines to the main text, on lines 134-139: “To extend the interaction to more complex lipid compositions, we tested the binding rate of MBP1 to a mixed lipid composition (DOPC/DPPC/Chol = 40:40:20). We found that the binding rate of MBP1 to DOPC/DPPC/Chol (40:40:20) vesicles is 0.018 nm/s (Fig. S1), similar to the rate observed for DOPC/Chol (70:30)

vesicles (0.017 nm/s). This consistency underscores the predominant role of Chol in enhancing binding kinetics, rather than the contributions from other lipid components”.

3. As mentioned in line 218: “FP1 and FP2 structures, obtained using AlphaFold2”, different software tools should be used to verify the prediction results of AlphaFold2 to ensure the accuracy of the initial structure used for simulation.

Authors: We appreciate the reviewer's comment. To verify the prediction results of AlphaFold2, we employed MD simulations in a water solution to assess the stability of the structures. This methodology is widely used and has proven effective in identifying potential mispredictions by AlphaFold. Furthermore, AlphaFold2 provides confidence intervals for its predictions. For the systems we studied, both peptides had a very high prediction score (above 90) for more than half of their sequences, particularly in the helical regions. The C-terminal regions of both structures were predicted with a lower confidence score, categorized as "OK" by AlphaFold2 (around 70). These regions were also observed to be more flexible in the simulations, which aligns with expectations for small membrane binding peptides. To clarify this point, the following text was added to the manuscript on lines 243-247: “The structures predicted by AlphaFold2 had pLDDT scores of 81.4 for MBP1 and 78.4 for MBP2, both of which fall within the 'confident' range according to AlphaFold(4). Notably, the alpha-helix regions exhibited even higher scores, exceeding 90, which is considered 'very high confidence' (**Fig. S5**). MBP1 and MBP2 structures were equilibrated using the MD protocol and inserted into the same two membrane environments as used previously.”

4. According to lines 221-226, for molecular dynamics simulations, observing protein insertion into the cell membrane depends on a variety of factors beyond simulation time, such as the accuracy of the model and the initial conditions set, like temperature, etc. (PLoS one 2012, 7, e47596). Therefore, the contents of this part are recommended to modify accordingly.

Authors: We appreciate the feedback. We have modified the sentence to: “Surprisingly, in our simulations, we did not observe the peptide fully intercalating into the lipid. In contrast, some other viruses, like flaviviruses, exhibit a behaviour where the membrane-binding peptide integrates into the membrane within a few nanoseconds of initial contact. Under our simulated conditions, a simulation time of 5 μ s should have been adequate to detect such insertion (13). To better understand the results, we calculated the average contact between the amino acids of the MBPs and the lipid membrane. We defined a contact as any amino acid atom within 4.5 Å of any membrane atom. The calculations performed every ns were averaged over 100 ns windows. The contact map reveals that MBP1 has a stronger preference for interacting with the cholesterol-enriched membrane as compared to the sphingomyelin membrane.” (see lines 251-260).

5. Based on the results in Fig. 3d, please explain the reason for the decrease in average contact in longer molecular dynamics simulations. Is this limited by the fact that the authors only used the FP structure for their simulations instead of the full-length Spike protein?

Authors: The decrease in average contact observed in longer molecular dynamics simulations, as shown in **Fig. 3d**, can be attributed to the fact that the MBP1 structures did not intercalate into the lipids. Instead, they remained on the lipid interface, moving across the membrane surface. This behavior is quite common for proteins interacting with membranes, as many proteins are restricted to "2D" movement along the membrane surface. As a result, contacts are continuously formed and broken over time. The darker colors in the plot indicate periods when the amino acids were more "buried" within the membrane. Additionally, the peptide lost most of its secondary structure while interacting with the membrane, as depicted in **Fig R7**.

Fig. R7. Cartoon show the secondary structure model of MBP1 interaction with lipid membrane. The left is top view, while right is side view.

6. Fig. 2 and Fig. 4 are ill-labeled and unsightly, which requires further revision.

Authors: Thank for this comment. We have revised the figure's label.

Reviewer #4 (Remarks to the Author):

The paper titled "Probing SARS-CoV-2 Fusion Domain-Membrane Interaction via Single-Molecule AFM-based Force Spectroscopy" investigates the binding between the SARS-CoV-2 spike protein's fusion peptide (FP) and host cell membranes. Employing both in vitro and computational approaches, the study reveals a distinct preference of the FP for cholesterol-rich membranes, emphasizing the crucial role of the internal disulfide bridge in stabilizing this interaction. This research offers significant insights into the mechanisms of viral membrane fusion, presenting potential targets for therapeutic intervention. Notably, the integration of single-molecule AFM-based force spectroscopy with computational simulations enhances the robustness of the findings.

Authors: We thank the reviewer for the positive assessment of our study.

Specific comments on data analysis of Figure 5:

The use of single-molecule force spectroscopy in constant-force mode to derive force-dependent lifetimes is meticulously analyzed using Bell's formula and its generalization (DHS model) under Kramers theory. This dual analytical method validates the findings, as shown by the comparable kinetic parameters between the two models. The execution of this comparison in Figure 5 is commendable, reinforcing the reliability of the results. However, further discussion on the scaling factor in the DHS model would be beneficial, particularly regarding how it affects the accuracy of the derived free-energy parameters. Addressing this would clarify the potential variability and enhance the robustness of the findings, thereby improving the overall impact of the study.

Authors: Thank you for your valuable assessment of our study and positive comments—particularly regarding the rigor for our experimental data and thermodynamic and kinetic model of single molecule force spectroscopy in constant force mode. The Dudko-Hummer-Szabo (DHS) model is an important theoretical framework used to describe the force-dependent dissociation kinetics of bonds, in the context of single-molecule force spectroscopy experiments. This model is extended from the classical Bell model by incorporating the concept of a multidimensional energy landscape and accounts for the non-linear relationship between force and the dissociation rate (1), which arises due to the non-linear shape of the energy barrier. In this regard, the scaling factor is essential for describing the shape of the energy barrier that a bond must overcome as force is applied. In our case, the scaling factor $a = 1$ corresponds specifies the nature of the underlying free-energy profile. This corresponds to a linear-cubic potential, which is similar to the behaviour assumed in the classical Bell model but adjusted to account for the curvature of the barrier. We have added this information to the main text (see lines 355-358): "By accounting for the complex nature of energy landscapes, the Dudko-Hummer-Szabo (DHS) model provides a more nuanced and accurate description of how molecular bonds behave under external force by considering the multidimensionality and shape of the energy barriers" and also added explanation in method section: "This corresponds to a linear-cubic potential, which is similar to the behavior assumed in the classical Bell model but adjusted to account for the curvature of the barrier (1)."

Minor corrections:

Fig 4c. Correct typo 'Bell modle Fit'

Authors: We have now have changed “Bell modle Fit” to “Bell model Fit” in the Fig 4c.

Legend of Fig 5b.

Pls specify the number of binding events analyzed to determine the frequencies.

Authors: We have now added the number of the specific binding events and the total number for analyzing frequencies in legend of Fig 5b.

Legend of Fig 5d.

- Explain colored data points, are they averages or mean? What are the error bars. Apparently the data has been binned, which bin has been applied?

Authors: Thank you for this comment. We have now re-written the legend of Fig 5d to provide additional context and explain the legend of Fig 5d.

“The bonds lifetime ($\Delta\tau$) of the MBP1 (treated with DMSO)-DOPC/Chol membrane bonds (gray circles) plotted against the binding force (ΔF , $n=237$). The colored circles show the average lifetime of the bonds determined for the different binding forces (Fig. S6). The lifetime of the bonds is classified with the average values for single (blue) and double (red) peptide-membrane bonds produced simultaneously. The error bars show the standard deviation (s.d.). The solid line shows the fit of the average bond lifetimes based on the DHS model.”

- The authors write 'Those data points were fitted using the DHS model.' Which data points the binned colored ones or the grey ones?

Authors: Those data points represent the binned lifetime (colored ones) that were fitted with the DHS model. We have now re-phrased this sentence: “The solid line shows the fit of the average bond lifetimes based on the DHS model.”

Legend of Fig 5f.

- The title of the legend needs correction. I think 'binds' needs to be replaced by 'binding'.

Authors: We have now have changed “binds” to “binding” in the legend of Fig 5.

- The authors write 'The kinetics and thermodynamics analysis reveal a binding' pls specify analysis of what?

Authors: We have now re-phrased this sentence: “The kinetics and thermodynamics analyses of the bond lifetimes in the MBP1 peptide (treated with DMSO)-DOPC/Chol membrane interaction reveals specific binding comprising of two distinct states in its binding free-energy landscape, which can be explained by a conformational change facilitated by the presence of the disulfide bridge.”

- The authors write 'can be explain' pls write 'can be explained'

Authors: We have now have corrected “can be explain” to “can be explained” in the legend of Fig 5f.

Reference:

1. O. K. Dudko, G. Hummer, A. Szabo, Theory, analysis, and interpretation of single-molecule force spectroscopy experiments. *Proceedings of the National Academy of Sciences* **105**, 15755-15760 (2008).
2. D. J. Müller *et al.*, Atomic Force Microscopy-Based Force Spectroscopy and Multiparametric Imaging of Biomolecular and Cellular Systems. *Chemical Reviews* **121**, 11701-11725 (2021).
3. C. J. Bustamante, Y. R. Chemla, S. Liu, M. D. Wang, Optical tweezers in single-molecule biophysics. *Nature Reviews Methods Primers* **1**, 25 (2021).
4. J. Jumper *et al.*, Highly accurate protein structure prediction with AlphaFold. *Nature* **596**, 583-589 (2021).
5. J. S. Low *et al.*, ACE2-binding exposes the SARS-CoV-2 fusion peptide to broadly neutralizing coronavirus antibodies. *Science* **377**, 735-742 (2022).
6. C. Dacon *et al.*, Broadly neutralizing antibodies target the coronavirus fusion peptide. *Science* **377**, 728-735 (2022).
7. W. Shi *et al.*, Cryo-EM structure of SARS-CoV-2 postfusion spike in membrane. *Nature* **619**, 403-409 (2023).
8. D. Birtles, J. Lee, Identifying Distinct Structural Features of the SARS-CoV-2 Spike Protein Fusion Domain Essential for Membrane Interaction. *Biochemistry* **60**, 2978-2986 (2021).
9. R. K. Koppiseti, Y. G. Fulcher, S. R. Van Doren, Fusion Peptide of SARS-CoV-2 Spike Rearranges into a Wedge Inserted in Bilayered Micelles. *Journal of the American Chemical Society* **143**, 13205-13211 (2021).
10. S. L. Schaefer, H. Jung, G. Hummer, Binding of SARS-CoV-2 Fusion Peptide to Host Endosome and Plasma Membrane. *The Journal of Physical Chemistry B* **125**, 7732-7741 (2021).
11. E. Durner, W. Ott, M. A. Nash, H. E. Gaub, Post-Translational Sortase-Mediated Attachment of High-Strength Force Spectroscopy Handles. *ACS Omega* **2**, 3064-3069 (2017).
12. F. Tian *et al.*, Verification of sortase for protein conjugation by single-molecule force spectroscopy and molecular dynamics simulations. *Chemical Communications* **56**, 3943-3946 (2020).
13. Y. S. Mendes *et al.*, The Structural Dynamics of the Flavivirus Fusion Peptide–Membrane Interaction. *PLOS ONE* **7**, e47596 (2012).

Point-by-Point Response to the Reviewers' Comments

Reviewer #1 (Remarks to the Author):

This manuscript is a resubmission and the authors have successfully addressed major concerns raised previously.

Authors: We are grateful that the reviewer recognized our efforts in addressing the major concerns raised in the initial round of reviews. Their comments have greatly contributed to improving the clarity and rigor of our work.

Reviewer #2 (Remarks to the Author)

I would like to thank the authors for their detailed responses and the additional experiments they have conducted. The revisions have addressed most of my concerns, and I am largely satisfied with the improvements made to the manuscript. However, there remains one major issue related to the S2 subunit that requires further clarification. Below are my specific comments:

1. The renaming of the fusion peptide region to "membrane-binding peptide" (MBP) is a sensible change given the recent cryo-EM data, and I appreciate the authors' effort to avoid confusion.

Authors: We appreciate your suggestion regarding the renaming of the peptide and are glad to hear that you support the change to "membrane-binding peptide" (MBP). We agree that this revision is crucial in light of the recent cryo-EM data, as it better reflects the peptide's function and avoids potential confusion.

2. The additional MD simulations and control experiments, particularly these involving the Arg846 mutation and MPP peptide, greatly strengthen the manuscript. However, I would suggest that structure prediction experiments also be performed on the Arg846 mutation to ensure that this mutation does not alter the peptide's structure. AlphaFold is a reasonable approach for such predictions, but the MBP region appears highly variable across different SARS-CoV-2 Spike protein variants. Moreover, the AlphaFold-predicted structure deviates slightly from published structures. To further solidify these findings, it would be valuable to confirm the structural integrity using experimental techniques such as HPLC if feasible within a reasonable timeframe.

Authors: Thanks for your encouraging feedback on the additional MD simulations and control experiments, particularly those involving the Arg846 mutation and MPP peptide. We appreciate your suggestion regarding structure prediction for the Arg846 mutation.

To clarify, the peptides in question were synthesized and their behavior was experimentally validated using AFM, confirming the initial results for MBP1 and MBP2. These experiments aligned well with our expectations and supported the conclusions drawn from the MD simulations.

To further address your concern, we used AlphaFold2 to predict the structural impact of the R846A mutation. As illustrated in the **Figures R1 & R2**, the mutation did not result in significant structural changes, with both the wild-type and mutant peptides displaying similar conformations (notably at position 32 where the mutation is located). This has been clarified in the manuscripts in lines 388-394.

"To further validate the critical role of the Arg846 residue, we synthesized a mutant version of MBP1 in which Arg846 was replaced by Gly (MBP1 mutant). Using AlphaFold2, we confirmed that both the wild-type MBP1 and the MBP1 mutant peptides exhibit similar overall

conformations, particularly at the site of the mutation. We then conducted AFM-based SMFS experiments using this point mutant peptide. Strikingly, the results showed that the MBP1 mutant failed to bind to the DOPC/Chol membrane under the tested conditions (Figs. S4,5), pointing out the critical role played by a single Arg846 residue. “

Figure R1: 3D structure rendering of the wild-type peptide (yellow) and its R846A mutant (orange), calculated using AlphaFold2. For clarity, Arg846 in the wild-type is depicted as sticks, while Ala846 in the mutant is represented in ball-and-stick form. The root-mean-square deviation (RMSD) between the two peptide backbones is 0.386 \AA^2 .

Figure R2. Predicted IDDT score per position, indicating the confidence of the structural prediction (scores range from 0 to 100, with higher values reflecting greater accuracy). The mutation is at position 32. Minimal differences are observed between the two plots, with the wild-type system on the left and the R846A mutant on the right.

However, it’s important to note that the peptide’s behavior in the membrane environment differs from the AlphaFold2-predicted structure. Our results indicate that the peptide strongly interacts with membrane surface polar head groups, undergoing conformational changes to optimize these interactions—details not captured by AlphaFold2’s predictions.

Additionally, the structural integrity of all peptides, including the Arg846 mutant, was confirmed by Genscript through rigorous quality control methods such as high-performance liquid chromatography (HPLC) and mass spectrometry (MS). These analyses validated both peptides purity and structural integrity (see lines 524-525).

Hence by thorough experimental validation, along with the MD simulations, we believe the evidence provided is sufficient to support the conclusions drawn in the manuscript.

3. While the authors have provided additional information regarding the S2 subunit used in their experiments, one key concern remains. The S2 subunit was purchased from a commercial source and is maintained in a soluble state. Without any modification to stabilize this subunit, it is highly likely that the S2 subunit remains in the post-fusion conformation. This point has not been sufficiently addressed in the manuscript, and the figure depicting the S2 subunit adds to the confusion. It raises concerns about the conclusions drawn from these experiments. If necessary, additional experiments or more detailed explanation should be provided.

Authors: Thank you for your valuable feedback regarding the S2 subunit used in our experiments. We would like to clarify that the S2 subunit was purchased from R&D Systems (ID: 10594-CV), a recombinant protein derived from human embryonic kidney (HEK 293) cells. This protein spans the region from Ser686 to Lys1211. Notably, this sequence encompasses a significant portion of the N-terminal region, which lies upstream of the S2 priming site cleaved by TMPRSS2, thus maintaining the subunit in its pre-fusion conformation.

In our experiments, we used AFM to investigate the interaction of the S2 subunit with lipid membranes. Initially, we did not observe any binding events, consistent with the S2 being in its pre-fusion state. After priming the S2 subunit with TMPRSS2, we repeated the AFM experiments and observed binding events that closely matched those obtained with the synthesized peptide. These results strongly suggest that the interaction observed reflects the N-terminal region of the primed S2 (S2') interacting with the lipid membrane.

Furthermore, we consider it unlikely that the S2 subunit transitions into the post-fusion conformation during these experiments. The S2 is grafted to the AFM tip prior to TMPRSS2 cleavage, and binding to the lipid membrane is monitored shortly after priming. The post-fusion transition involves insertion of the fusion peptide into the membrane, followed by a refolding into a hairpin-like structure. However, in our experiments, the binding behavior remained consistent with that observed using the MBP1 peptide, which lacks the fusion peptide, indicating that no conformational shift to the post-fusion state occurred.

We hope this explanation addresses your concern. Given that the binding interactions observed after TMPRSS2 activation are consistent and align with the synthesized peptide data, we believe the conclusions drawn from these experiments are robust. This is clarified in the manuscript in lines 272-308.

4. The new control experiments on cholesterol depletion and repletion provide strong support for the conclusions. I appreciate the clarity of the results showing that viral infection increases following cholesterol repletion. It would be more helpful to clarify the potential role of the actual fusion peptide during these experiments.

Authors: Thank you for your encouraging feedback on the cholesterol depletion and repletion experiments.

Regarding your comment on the potential role of the fusion peptide, we would like to clarify that the primary focus of this work is on how the membrane binding peptide (MBP) within the S2 domain interacts with the membrane, in particular the influence of cholesterol on this interaction. Our in vitro data, combined with the infectivity assays, show that cholesterol plays an important role in the viral entry process, specifically during the binding of the S2 MBP to the membrane- a step that is physiologically relevant for viral entry. These findings suggest that the interaction between the S2 domain and cholesterol enhances viral attachment to target cell membranes. While cholesterol could potentially influence later steps in the fusion process, exploring its role in membrane fusion itself is beyond the scope of this study.

To clarify this in the manuscript, we have added the following sentence (lines 440-445): " These infectivity assays demonstrate that cholesterol is essential for efficient viral binding and entry, likely by increasing the binding affinity of viral membrane-binding peptide to host membrane, consistent with our in vitro findings that the MBP within the S2 domain interacts with cholesterol after TMPRSS2 priming to facilitate membrane attachment. Cholesterol levels may also influence subsequent steps such as membrane fusion."

5. The authors' observations regarding the interactions between MBP1 and the membrane are intriguing and open new avenues for future research. I appreciate the authors' consideration of this point.

Authors: Thank you.

In conclusion, I am largely satisfied with the revisions, but the issue concerning the S2 subunit's structural state remains unresolved. I would recommend addressing this point in detail to ensure the robustness of the conclusions drawn from the S2 subunit experiments. Once this is resolved, I believe the manuscript will be suitable for publication.

Authors: Thank you very much. As explained above, we have now clarified this point in the manuscript. We explicitly addressed the structural considerations of the S2 subunit in the revised manuscript, including a detailed explanation of how the experimental setup ensures that we are probing the biologically relevant form of S2 subunit.

We hope this additional clarification resolves the concern, and we appreciate your constructive input throughout the review process.

Reviewer #3 (Remarks to the Author)

In this round, the authors have implemented a series of revisions and responses to the previous review comments.

However, despite the authors' efforts in addressing the review comments, the fifth point in Minor comments has not received sufficient attention. I have concerns about the role of the Spike protein S2 subunit in the virus-host cell membrane fusion process. Although the manuscript primarily focuses on the interaction between the membrane-binding peptide (MBP) and the lipid membrane, the effect of S2 on cell membrane fusion following TMPRSS2 cleavage should not be overlooked, whether in vitro or in silico experiments.

Authors: We appreciate your insightful comments regarding the role of the Spike protein's S2 subunit in the virus-host membrane fusion process, particularly following TMPRSS2 cleavage. While our manuscript primarily focuses on the interaction between the membrane-binding peptide (MBP) and the lipid membrane, we agree that the S2 subunit plays a critical role in mediating membrane fusion post-cleavage.

In response to your concern, we have reviewed recent studies, including a recent 2024 Science paper, which provides a comprehensive view of how the S2 subunit facilitates membrane fusion following activation by TMPRSS2 ¹. This study uses high-resolution cryo-EM to capture key intermediate stages in the S2 refolding process, demonstrating how the S2 subunit transitions from its prefusion to postfusion states, ultimately driving membrane fusion. The conformational changes observed in the S2 subunit, particularly the formation of the six-helical bundle (6-HB), are essential for the merger of viral and host membranes.

In addition, both the studies by Huang et al. ² and Fraser et al. ³ emphasize the importance of the S2 subunit in conjunction with TMPRSS2 cleavage. TMPRSS2 mediates the critical cleavage of the S1/S2 site, which is necessary for exposing the S2 fusion machinery, enabling it to form the HR1-HR2 complex and execute membrane fusion. While our focus was on the MBP-lipid interaction, we recognize that the dynamic role of the S2 subunit post-cleavage is a key aspect that requires further study.

We have expanded our discussion to include this essential role of the S2 subunit and TMPRSS2 cleavage and plan to address these dynamics in future experiments and simulations (Lines 492-502).

“Beyond the MBP-lipid interaction analyzed in this study, the spike protein's S2 subunit plays an equally crucial role in mediating the virus-host membrane fusion process. Following cleavage by TMPRSS2, the S2 subunit undergoes substantial conformational changes, transitioning from its prefusion to postfusion states, as described in recent high-resolution cryo-EM studies. These changes lead to the formation of the six-helical bundle, which is essential for bringing the viral and host membranes into close proximity and enabling fusion. TMPRSS2-mediated cleavage has been shown to be vital for the activation of the S2 subunit and its role in fusion. While our work has focused on the MBP-lipid membrane interaction, the S2 subunit's dynamics post-cleavage also warrant further investigation, as they are integral to the fusion

process. Future studies could explore both the in vitro and in silico dynamics of the S2 post-cleavage transition to provide a more holistic view of membrane fusion.”

In addition, some of the figures are still not satisfactory and need further revision.

Authors: Thank you for your comment. We have carefully reviewed all the figures again and made several changes to improve their clarity and presentation.

Reviewer #4 (Remarks to the Author):

The authors have addressed all of my comments satisfactorily. The revised manuscript has been much improved and is a pleasure to read.

Authors: Thank you.

References:

- 1 Grunst, M. W. *et al.* Structure and inhibition of SARS-CoV-2 spike refolding in membranes. *Science* **385**, 757-765, (2024).
- 2 Huang, Y., Yang, C., Xu, X.-f., Xu, W. & Liu, S.-w. Structural and functional properties of SARS-CoV-2 spike protein: potential antivirus drug development for COVID-19. *Acta Pharmacol. Sin.* **41**, 1141-1149, (2020).
- 3 Fraser, B. J. *et al.* Structure and activity of human TMPRSS2 protease implicated in SARS-CoV-2 activation. *Nat. Chem. Biol.* **18**, 963-971, (2022).